# Robot-R1: Reinforcement Learning for Enhanced Embodied Reasoning in Robotics

**Dongyoung Kim**[1,4], **Sumin Park**[1] , **Huiwon Jang**[1], **Jinwoo Shin**[1,4]
**Jaehyung Kim**[*2], **Younggyo Seo**[*3]
[1]KAIST, [2]Yonsei University, [3]UC Berkeley, [4]RLWRLD
kingdy2002@kaist.ac.kr

## Abstract

Large Vision-Language Models (LVLMs) have recently shown great promise in advancing robotics by combining embodied reasoning with robot control. A common approach involves training on embodied reasoning tasks related to robot control using Supervised Fine-Tuning (SFT). However, SFT datasets are often heuristically constructed and not explicitly optimized for improving robot control. Furthermore, SFT often leads to issues such as catastrophic forgetting and reduced generalization performance. To address these limitations, we introduce ROBOT-R1, a novel framework that leverages reinforcement learning to enhance embodied reasoning specifically for robot control. ROBOT-R1 learns to predict the next keypoint state required for task completion, conditioned on the current scene image and environment metadata derived from expert demonstrations. Inspired by the DeepSeek-R1 learning approach, ROBOT-R1 samples reasoning-based responses and reinforces those that lead to more accurate predictions. To rigorously evaluate ROBOT-R1, we also introduce a new benchmark that demands the diverse embodied reasoning capabilities for the task. Our experiments show that models trained with Robot-R1 outperform SFT methods on embodied reasoning tasks. Despite having only 7B parameters, ROBOT-R1 even surpasses GPT-4o on reasoning tasks related to low-level action control, such as spatial and movement reasoning.

## 1 Introduction

Recently, Large Vision-Language Models (LVLMs) have shown significant promise in robotics [1, 2]. By jointly processing visual inputs and natural language, LVLMs offer a powerful interface for interpreting complex real-world scenarios and enabling high-level reasoning on robot control [3–6]. Specifically, they have been employed in robotics by providing high-level actions in the form of textual descriptions [7] and latent representations [8, 9] to generate low-level actions, or by directly generating low-level actions [10]. These capabilities not only improve the performance and generalization of robotic systems but also enhance the human-robot interaction interface by enabling intuitive, language-driven control.

Despite these advances, LVLMs often struggle to translate their general commonsense into the nuanced embodied reasoning required for controlling robots. For example, they often fail to accurately understand the spatial relationships crucial for low-level control [11] or fail to generate high-level plans [12]. Consequently, achieving high performance in robotic tasks typically requires additional training in domain-specific data. A common approach involves Supervised Fine-Tuning (SFT) using question-answering datasets that pair task instructions with descriptions of various embodied reasoning types, such as action planning and spatial reasoning [13–15]. However, embodied reasoning

---

*Equal advising

39th Conference on Neural Information Processing Systems (NeurIPS 2025).

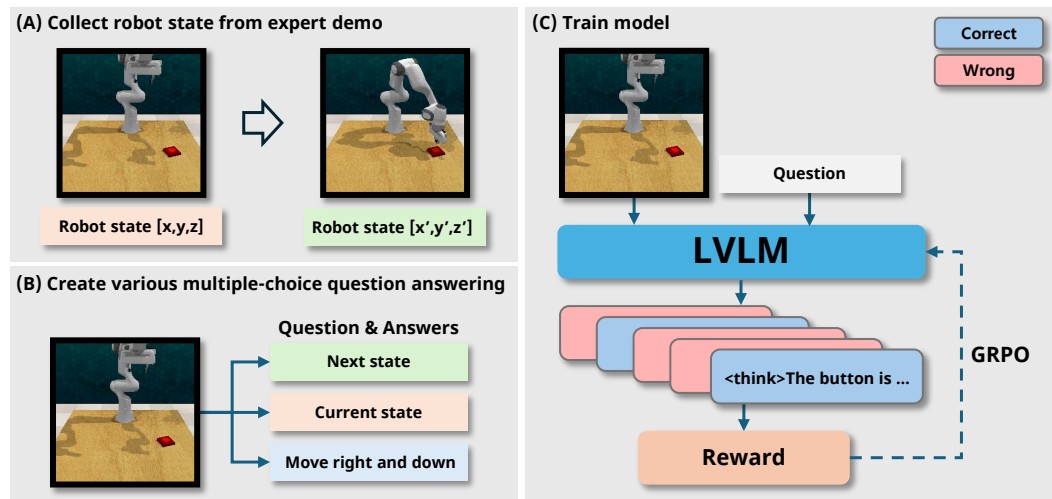

Figure 1: **Illustration of the ROBOT-R1 framework.** (a) ROBOT-R1 uses robot states and image observations from expert demonstrations to create a dataset. (b) These data are reformulated into three different multiple-choice question answering (MCQA) tasks: predicting next states, current states, and movements. (c) During training, an LVLM solves MCQA tasks with reasoning which is then optimized using the GRPO algorithm [17] to reinforce reasoning pathways.

tasks for SFT are often heuristically designed and thus not fully optimized for actual robot action prediction; for instance, the linguistic expressions in SFT datasets often fail to capture the precise quantitative details essential for low-level robot control. Moreover, models trained via SFT often struggle with input and output formats they were not trained on (*i.e.*, distribution shifts), which leads to catastrophic forgetting that degrades previously acquired knowledge, including general conversational abilities [16].

Meanwhile, recent advances led by Deepseek-R1 [17] have demonstrated the effectiveness of reinforcement learning (RL) in eliciting and reinforcing reasoning pathways, often achieving superior performance and generalization compared to SFT methods. Inspired by this, we introduce ROBOT-R1, a novel framework that employs RL to effectively train LVLMs with embodied reasoning capabilities tailored specifically for robotic control (see Figure 1).

The key idea of ROBOT-R1 is to train LVLMs to predict the next keypoint state necessary for task completion through a reasoning process, and to optimize this reasoning via RL to maximize prediction accuracy. However, since the keypoint lies in a continuous space, the action space is vast, making exploration particularly challenging. This makes it difficult for the model to efficiently learn reasoning strategies through trial and error. To address this, Robot-R1 reformulates the next-state prediction problem as a multiple-choice question-answering (QA) problem. This discrete formulation narrows the action space, making the learning process more efficient. To further enhance state understanding, we add two auxiliary QA tasks: (i) current state prediction QA, where the model predicts the robot's current state from visual observations, and (2) movement prediction QA, where the model predicts rule-based linguistic descriptions of state changes (e.g., "move up", "move down").

To analyze how ROBOT-R1 improves embodied reasoning capabilities in trained models, we design a novel benchmark called the ROBOT-R1 Bench (see Figure 2). Through this benchmark, we observe that models trained with ROBOT-R1 achieve over a 28% improvement in embodied reasoning tailored for low-level action control. In particular, despite having only the 7B parameters, ROBOT-R1 outperforms several major commercial models, including GPT-4o [18], in this domain. To evaluate whether the embodied reasoning abilities acquired through ROBOT-R1 transfer effectively to other tasks, we conduct evaluations on two external benchmarks. First, on EmbodiedBench Manipulation [19], a vision-driven robot agent benchmark, ROBOT-R1 yields a 31% improvement in task performance. Moreover, on SpatialRGPTbench [20], which tests 3D spatial reasoning, name achieves approximately 40% improvement in quantitative metrics and about 60% improvement in qualitative metrics. These results contrast with conventional SFT approaches, which show limited improvement in embodied reasoning and tend to exhibit performance degradation when applied to out-of-distribution tasks. This suggests that ROBOT-R1 can effectively learn generalizable and diverse embodied reasoning capabilities simply by next-state prediction.

## 2 Related Work

**Embodied reasoning for robot control.** Integrating Large Vision-Language Models (LVLMs) into robotic systems has recently emerged as a promising direction [1, 2]. These models leverage their vision-language understanding capabilities to interpret task instructions and process visual observations. Based on this understanding, LVLMs have been used to generate low-level actions directly [10, 21] or to produce high-level action [3–7, 12, 22]. Recent efforts have focused on enhancing the embodied reasoning capabilities of LVLMs to improve performance in complex long-horizon tasks [23, 24]. For instance, their ability to reason over language and visual inputs has been exploited for action planning, high-level action prediction, and spatial reasoning [25–27]. A common approach involves constructing embodied reasoning question-answering (QA) datasets, which are designed to aid robot control [15, 11, 28–31]. These datasets are then used to fine-tune models via supervised fine-tuning (SFT). However, a notable limitation is that such embodied reasoning datasets are often heuristically constructed and may not be explicitly optimized for robot control.

**Reinforcement learning for encourage reasoning.** In large language models (LLMs), generating intermediate reasoning steps – commonly known as Chain-of-Thought (CoT) reasoning – has proven to be an effective strategy for improving performance across a wide range of tasks [32]. This success has motivated substantial research into enhancing reasoning abilities through various ways, such as prompting techniques or the distillation of reasoning paths from stronger models [33–36]. More recently, reinforcement learning (RL) has emerged as a strong alternative to SFT for training reasoning models. In particular, research such as DeepSeek-R1 [17] has introduced RL-based frameworks where the model first generates a reasoning trace, then produces an answer based on this trace, and is optimized using reward signals based on answer correctness. This RL-based approach not only improves reasoning quality but also enhances sample efficiency and generalization across a variety of tasks [37–39]. Consequently, it has been successfully applied to domains such as mathematics [40, 41], agentic tasks [42, 43], and even vision-related problems [44, 45], yielding substantial performance gains. These advances suggest that applying RL to train embodied reasoning for robot control could effectively address key limitations of SFT-based approaches.

## 3 Method

In this section, we present the components in the following order:

- ○ **Preliminaries**: Definition of robot state, and GRPO [46] algorithm for RL training.
- ○ **Data Generation**: How to prepare training dataset for ROBOT-R1.
- ○ **Multiple-choice QA Base Training**: How to train LVLM via RL with the generated datasets.
- ○ **ROBOT-R1 Bench**: Introduce a new evaluation benchmark for embodied reasoning that assesses various reasoning capabilities (*e.g.* spatial understanding) in robotic control scenarios.

### 3.1 Preliminaries

**Robot state.** In this work, we utilize a Franka Panda robot whose state is represented as a 7-dimensional vector consisting of the end-effector's 3D cartesian position $(x, y, z)$, orientation (roll, pitch, yaw), and a binary gripper state (open/closed). For simplicity, following Liang et al. [22], we consider only the Cartesian position (x, y, z) of the end-effector as the robot state $s$.

**Problem setup.** We denote the LLM policy as $\pi_\theta$ that generates an answer $o \sim \pi_\theta(q)$ in response to a question $q$. A reward $r = f_{\texttt{reward}}(o)$ is obtained by evaluating the generated answer through a pre-defined reward model. This reward model is designed to assign higher values to answers that more accurately predict the next state. The LLM policy is then optimized to maximize this reward.

**Group relative policy optimization.** We employ the Group Relative Policy Optimization (GRPO) algorithm [17] to optimize the policy. Specifically, for a given query $q$, we generate $G$ different responses $\mathbf{o} = \{o_1, o_2, o_3, , , o_G\}$ from the current policy $\pi_{\theta_{\text{old}}}$. Each response $o_i$ is evaluated using the reward model to produce a corresponding reward $r_i$, resulting in the set $\mathbf{r} = \{r_1, r_2, ... r_G\}$. For each response, we compute an advantage score $A_i$, using the mean and standard deviation of the rewards for that query, *i.e.*, $A_i = \frac{r_i - \text{mean}(\mathbf{r})}{\text{std}(\mathbf{r})}$. Finally, GRPO updates the policy by maximizing this

advantage while applying a KL penalty, through the following objective:

$$\mathbb{J}_{\text{GRPO}}(\theta) = \mathbb{E}_{q \sim P(Q), \{o_i\}_{i=1}^{G} \sim \pi_{\text{old}}(Q|q)} \Bigg[$$

$$\frac{1}{G} \sum_{i=1}^{G} \min \left( \frac{\pi_\theta(o_i|q)}{\pi_{\theta_{\text{old}}}(o_i|q)} A_i, \text{clip} \left( \frac{\pi_\theta(o_i|q)}{\pi_{\theta_{\text{old}}}(o_i|q)}, 1 - \varepsilon, 1 + \varepsilon \right) A_i \right) - \beta \mathbb{D}_{\text{KL}}(\pi_\theta \| \pi_{\text{ref}}) \Bigg],$$

where $\varepsilon$ and $\beta$ are hyperparameters.

### 3.2 Data Generation

For training ROBOT-R1, we use multiple-choice question answering (QA) data, which includes a primary task of predicting next waypoint (*i.e.*, keypoint), along with two auxiliary tasks: (i) estimating the current state from an image and (ii) determining the necessary movements to reach the next waypoint. These tasks are constructed using information extracted from expert demonstrations.

**MetaData extraction.** Without any contextual information, LVLMs struggle to infer low-level states. To address this, we utilize metadata $M$, which encodes details about the current task and the robot's low-level state, and use it to generate question prompts. The metadata $M$ encompasses three essential types of information. First, it includes details about fixed reference points within the robot's environment, such as the center of a table in table-top manipulation scenarios. Second, it defines the 3D coordinate system, including the positive direction of each axis, which helps interpret how spatial changes are reflected in numerical state changes. Third, it incorporates the dimensions ($x$, $y$, $z$) of a consistently present object, such as the robot's end-effector, to provide a reference for scale estimation. This metadata $M$ is subsequently used as conditional input during question generation.

**Waypoint prediction QA.** The goal of this task is to predict a future waypoint state. We represent a demonstration as a sequence of frames $D = \{(o_0, s_0), (o_1, s_1), \ldots, (o_N, s_N)\}$, where each frame consists of an observation $o_t$ and its corresponding robot state $s_t$. A subset of these frames is designated as *keypoints*, denoted by indices $K = \{k_0, k_1, \ldots, k_M\}$, which mark significant changes in the trajectory or completion of crucial subgoals [47]. The task is to predict the state $s_{k^*}$ of the next keypoint following the current time step $t$, where $k^* = \min\{k_j \in K \mid k_j > t\}$. Each training example is thus represented as a tuple $(s_t, o_t, s_{k^*})$. To make a multiple-choice question, we randomly sample three distractor states $s_{d1}, s_{d2}, s_{d3}$ from the robot's valid state space. These candidates, along with the correct next waypoint $s_{k^*}$, are shuffled to create the final question input $Q_{\text{waypoint}}(M, s_t, o_t, \text{shuffle}(\{s_{k^*}, s_{d1}, s_{d2}, s_{d3}\}))$ where the correct answer is $A = s_{k^*}$.

**Current state prediction QA.** The goal is to identify the correct current state $s_t$. The question prompt includes the metadata $M$ and the current visual observation $o_t$. To construct a multiple-choice question, we randomly sample three distractor states $\{s'_{d1}, s'_{d2}, s'_{d3}\}$ from the state space. The question is formulated as $Q_{\text{state}}(M, o_t, \text{shuffle}(\{s_t, s'_{d1}, s'_{d2}, s'_{d3}\}))$ where the answer is $A = s_t$.

**Movement prediction QA.** The goal of this task is to predict the movement (*e.g.*, "move up", "move right") that the robot should take to move from the current state toward the next waypoint. To extract labels for these movement, we apply rule-based heuristics based on the change in 3D Cartesian space. Specifically, we compute the difference in position between the current and next waypoint states, $s_{k^*} - s_t$, and identify the primary direction of movement along the $x$, $y$, and $z$ axes. If there is a non-zero change along an axis, a corresponding directional command is generated (*e.g.*, "move forward" for a positive change in the x-axis); this label is denoted as $a_t$. If the change along an axis is less than or equal to half of the largest change across all axes, the adverb "slightly" is added (*e.g.*, "Slightly move backward"). To construct a multiple-choice question, we randomly sample three adjacent movement $= \{a'_{d1}, a'_{d2}, a'_{d3}\}$. The question is formulated as $Q_{\text{movement}}(M, o_t, s_t, \text{shuffle}(\{a_t, a'_{d1}, a'_{d2}, a'_{d3}\}))$ where the answer is $A = a_t$.

### 3.3 Multiple-choice QA Base Training

Following DeepSeek-R1-Zero [17], we train LVLMs to explicitly generate their reasoning process. During training, the policy is instructed to output its thought process enclosed within `<think>` and `</think>` tags, followed by the final answer enclosed within `<answer>` and `</answer>` tags. The

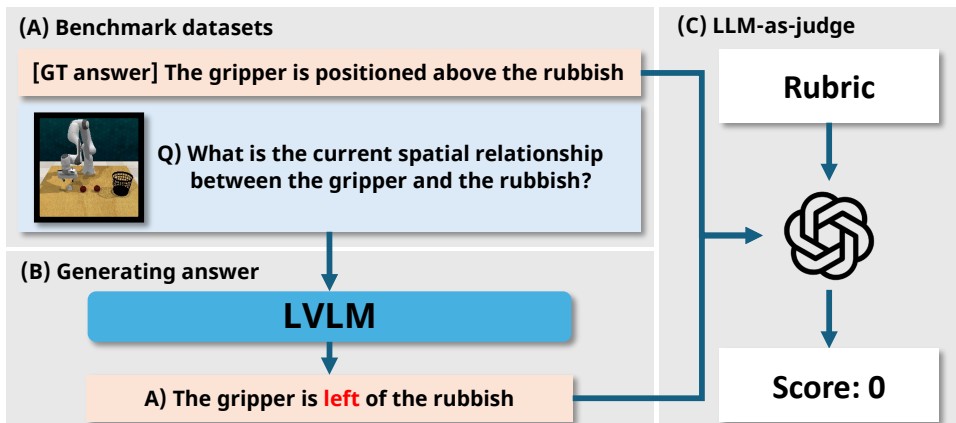

Figure 2: **Illustration of the ROBOT-R1 Bench.** (a) ROBOT-R1 Bench consists of human-written questions paired with corresponding ground truth (reference) answers. (b) The LVLM under evaluation takes each question along with its associated image as input and generates an answer. (c) The generated answers are scored using GPT-4o, based on predefined rubrics and ground truth answers.

Table 1: **Example of the ROBOT-R1 Bench question and response.**

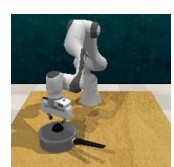

**Question:** What is the immediate next meaningful subtask (i.e., a single step within a larger sequence) that should be performed given the current scene?

**Model Answer :** `<think>`robot appears to be positioned above a saucepan with a lid. The task is to take the lid off the saucepan. The immediate next step would involve moving the gripper towards the lid to grasp it. `</think>` `<answer>`the gripper towards the lid of the saucepan.`</answer>`

policy is optimized using GRPO. The reward signal $R$ for RL consists of two components: a format reward $r_f$ and an answer correctness reward $r_a$, combined as $R = r_f + r_a$. The format reward $r_f$ encourages adherence to the specified output structure. The answer reward $r_a$ provides rule-based positive feedback if the model's answer, within `<answer>...</answer>`, exactly matches the correct option in the multiple-choice QA problem.

### 3.4 ROBOT-R1 Bench: A Novel Benchmark for Evaluating Embodied Reasoning

Existing visual question answering (VQA) benchmarks for embodied reasoning primarily focus on general visual understanding, without explicitly evaluating the nuanced reasoning processes behind robotic behavior [48–50]. Meanwhile, robotics-focused benchmarks often rely on simulation-based evaluation [19, 51, 52], priortize task successes as the main metric, or focus on tasks that are not aligned with the table-top manipulation scenarios central to our work. While these benchmarks provide valuable insights, they do not directly assess a model's ability to reason about robot actions. To address this gap, we introduce ROBOT-R1 Bench, a new benchmark designed to evaluate embodied reasoning through open-ended answering grounded in robot demonstrations.

**Design features of ROBOT-R1 Bench.** In practical applications, models typically face open-ended decisions rather than selecting from pre-defined options. To better reflect this, ROBOT-R1 Bench adopts an open-ended QA format that more closely resembles practical use cases. All questions are based on images from expert demonstrations, ensuring visually grounded and realistic robotic scenarios. ROBOT-R1 Bench supports fine-grained evaluation across four key reasoning types: *planning, high-level action reasoning, movement reasoning, and spatial reasoning.* Each question is crafted to assess one of these abilities, allowing for evaluation across both high-level decision making (planning and high-level action reasoning) and low-level control reasoning (movement reasoning and spatial reasoning).

**Dataset construction.** The ROBOT-R1 Bench dataset consists of 10 tasks from RLBench [53]. For each task, we randomly sample five frames from expert demonstratrions, resulting in a total of 50

Table 3: **ROBOT-R1 Bench results.** Performance on embodied reasoning tasks tailored for low-level control evaluated using the ROBOT-R1 Bench. ROBOT-R1 achieves the highest overall performance, outperforming leading commercial models across benchmark scores.

| Model | Movement | | | Spatial | | |
|---|---|---|---|---|---|---|
| | In | Out | Avg | In | Out | Avg |
| GPT-4o-mini [18] | 0.64 | **0.56** | 0.60 | 1.73 | 1.14 | 1.48 |
| GPT-4o [18] | 0.92 | 0.52 | 0.72 | 1.70 | 1.07 | 1.43 |
| Claude-3-Opus [59] | 0.40 | 0.24 | 0.32 | 0.97 | 0.43 | 0.74 |
| Claude-3.5-Haiku [60] | 0.68 | 0.48 | 0.58 | 1.41 | 0.71 | 1.11 |
| Claude-3.5-Sonnet-v2 [61] | 0.96 | 0.28 | 0.62 | 1.49 | 0.75 | 1.17 |
| Claude-3.7-Sonnet [57] | 0.76 | 0.48 | 0.62 | 1.65 | 1.07 | 1.4 |
| Gemini-1.5-Flash [62] | 0.60 | 0.16 | 0.38 | 1.57 | 0.93 | 1.29 |
| Gemini-1.5-Pro [62] | 0.76 | 0.40 | 0.58 | 1.49 | 1.14 | 1.34 |
| Gemini-2.0-Flash [58] | 0.52 | 0.40 | 0.46 | 1.76 | 1.14 | 1.49 |
| Qwen2.5-VL-7B-Ins [56] | 0.64 | 0.52 | 0.58 | 1.62 | 1.11 | 1.40 |
| w/ Direct SFT | 0.12 | 0 | 0.06 | 0.08 | 0.04 | 0.06 |
| w/ CoT SFT | 0.84 | **0.56** | 0.70 | 0.46 | 0.07 | 0.29 |
| w/ **ROBOT-R1 (Ours)** | **0.96** | **0.56** | **0.76** | **1.76** | **1.18** | **1.51** |

images. For each image, experienced researchers manually created questions and detailed reference answers, targeting the four key reasoning abilities. The final dataset consists of 215 open-ended questions in total: 65 for spatial reasoning and 50 for each of the other three reasoning types.

**Evaluation.** Model responses are evaluated against reference answers. For consistent and objective scoring, we adopt an "LLM-as-judge" approach using GPT-4o [18, 54, 55]. Based on a predefined rubric tailored to each reasoning type, GPT-4o assesses the model's answers for accuracy, logical coherence, and completeness, and assigns a numeric score in $[0, 3]$. This allows for fine-grained quantitative analysis of the model's embodied reasoning capabilities (See Figure 2 and Table 1).

## 4 Experiment

This section details the experimental setup for training ROBOT-R1, and the evaluation results. Our goal is to validate the effectiveness of ROBOT-R1 in learning embodied reasoning for robot manipulation tasks. We utilize the Qwen2.5-7b-VL-Ins [56] as the base model for all experiments.

### 4.1 Experimental Setup

**Training data.** The training data used in our experiments are generated using the built-in data generator in RLBench [53]. We collect 50 demonstrations per task from the `variation 0` settings, rendering them at 224 × 224 resolution using OpenGL3. Waypoints from these demonstrations are extracted using the waypoint extraction functionality provided in the ARM [47] repository[1]. The tasks selected from RLBench for training our model include: `pick_up_cup`, `push_button`, `put_rubbish_in_bin`, `phone_on_base`, and `take_lid_off_saucepan`. For generating the waypoint prediction QA training data, as described in Section 3,

Table 2: **Summary of ROBOT-R1 Bench results.** In embodied reasoning tailored for high-level control, ROBOT-R1 significantly outperforms SFT, the baseline training method.

| Model | Planning | High-level Action |
|---|---|---|
| GPT-4o [18] | **1.96** | **2.02** |
| cluade-3-7-sonnet [57] | 1.72 | 1.58 |
| gemini-2.0-flash [58] | 1.84 | 1.12 |
| Qwen2.5-VL-7B-Ins [56] | 1.66 | 1.04 |
| w/ Direct SFT | 0 | 0.04 |
| w/ CoT SFT | 0.60 | 1.28 |
| w/ **ROBOT-R1 (Ours)** | 1.44 | 1.30 |

a frame extraction interval of $t = 10$ was used between the current frame $(o_t, s_t)$ and the selected future keypoint frame $(s_{k^*})$, similar to demonstration augmentation proposed in ARM. Consequently, each task contains approximately 2.5K questions, resulting in a total of around 7.5K QA pairs across the three QA tasks used for training.

---

[1] https://github.com/stepjam/ARM

**Baseline models.** The proposed ROBOT-R1 learns to predict the next waypoint state from expert demonstrations by generating an explicit reasoning process. To evaluate the importance of this learned reasoning, we establish two baseline models trained via Supervised Fine-tuning (SFT):

○ **Direct SFT**: This model is fine-tuned on a QA dataset to directly predict the next waypoint state ($s_{k^*}$) from the current observation ($o_t$) and state ($s_t$), without any intermediate reasoning steps. This baseline helps assess the performance gain achieved by incorporating reasoning.

○ **CoT SFT**: This model is fine-tuned on a QA dataset where the input prompt includes a manually structured Chain-of-Thought (CoT) path. This CoT path sequentially incorporates planning, high-level action reasoning, and movement components to guide the prediction of the next waypoint state, similar to Zawalski et al. [15]. This baseline enables comparison between the reasoning learned via ROBOT-R1 against reasoning guided by a predefined, structured thought process.

By comparing ROBOT-R1 against these baselines, we aim to highlight the importance of (i) incorporating reasoning at all (v.s. Direct SFT) and the adaptiveness of reasoning learned through reinforcement learning (v.s. CoT SFT). See Appendix B for details.

**Implementation detail.** ROBOT-R1 is trained using GRPO as described in Section 3. We use the hyperparameter configurations in the EasyR1 workspace.[2] For the training process, we utilize a batch size of 128 over a 5 epoch. During GRPO updates, 5 samples were generated per prompt with a sampling temperature of 1.0. The rollout batch size is set to 512. We use a learning rate of $1.0 \times 10^{-6}$ with a weight decay of $1.0 \times 10^{-2}$. For the SFT baselines, we use the same batch size, but learning rate is $1.0 \times 10^{-5}$. We use the hyperparameters provided in the Qwen2-VL-Finetune workspace.[3]

### 4.2 Robot-R1 Bench

**Setup.** To reduce variance in benchmark results, all models generate responses with a temperature setting of 0. To ensure the robustness of our benchmark and allow for relative performance comparisons, we evaluate the performance of several widely used commercial models across different versions, including GPT [18], Claude [57, 59–61], and Gemini [58, 62, 63]. Furthermore, we evaluate the performance of our own models: the base Qwen2.5-VL-7B-Instruct model [56], its variant fine-tuned via SFT, and the model trained using our proposed ROBOT-R1 method (see Appendix B for details).

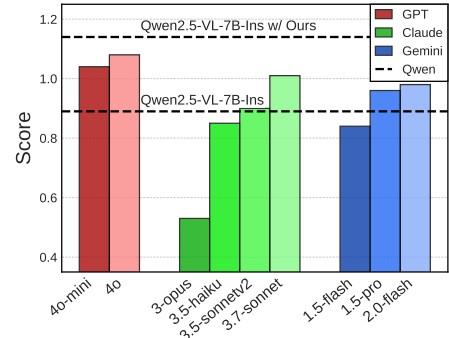

Figure 3: **ROBOT-R1 Bench results.** In embodied reasoning tailored for low-level control, ROBOT-R1 outperforms all previously reported models.

**Results.** Table 3 presents the results across the reasoning benchmarks focused on low-level control, and Figure 3 provides its visualization to help understanding. We find that commercial model performance improves consistently with their version updates, indicating the benchmark's sensitivity to model quality. Notably, the ROBOT-R1-trained model achieves the highest performance across both the overall benchmark and its individual components, outperforming all commercial models evaluated. Specifically, in movement prediction, our method outperforms even the SFT-trained model, which is explicitly trained to predict accurate actions. Table 2 further show the performance in reasoning tasks tailored for high-level control. Here, the correlation between model version and benchmark performance is less pronounced. We hypothesize that this is due to the inherently high variability in valid high-level actions for robotic control. Nevertheless, we still observe a general trend of increasing benchmark performance as model quality improves. Importantly, our ROBOT-R1 exhibits significant performance gains in high-level action reasoning compared to the base and the SFT-trained model. This suggests that effective high-level reasoning can emerge from training exclusively on low-level control information, even without explicit high-level action supervision. Interestingly, we find that performance on planning slightly decreases, likely because our training objective focuses primarily on next keypoint prediction rather than long-horizon planning.

---

[2]https://github.com/hiyouga/EasyR1
[3]https://github.com/2U1/Qwen2-VL-Finetune

Table 4: **EmbodiedBench Manipulation results.** Success rate (%) of ROBOT-R1, SFT baselines, and various commercial models in evaluating the performance of EmbodiedBench Manipulation [19], which evaluates LVLM performance for vision-based robotic agents.

| Model | Base | Common | Complex | Spatial | Visual | Avg. |
|---|---|---|---|---|---|---|
| GPT-4o-mini [18] | 4.2 | 6.3 | 2.1 | 10.4 | 0 | 4.6 |
| Cluade-3.5-Haiku [59] | 12.5 | 12.5 | 12.5 | 16.7 | 13.9 | 13.6 |
| Gemini-1.5-flash [63] | 14.6 | 10.4 | 4.2 | 10.4 | 8.3 | 9.6 |
| Qwen2.5-VL-7B-Ins [56] | 6.3 | 6.3 | **6.3** | **14.6** | 11.1 | 8.92 |
| w Direct SFT | 0 | 0 | 0 | 0 | 0 | 0 |
| w CoT SFT | 0 | 0 | 0 | 0 | 0 | 0 |
| w **ROBOT-R1(Ours)** | **12.5** | **8.3** | **6.3** | **14.6** | **16.7** | **11.68** |

## 4.3 EmbodiedBench Manipulation

Next, to evaluate whether ROBOT-R1 not only enhances embodied reasoning capabilities but also leads to improved performance in an actual robotic agent environment, we consider the EmbodiedBench Manipulation benchmark [19], a vision-driven agent assessment platform built upon the RLBench simulation environment [53]. This benchmark evaluates a model's ability to complete manipulation tasks by predicting low-level actions (a 7-dimension action vector) using in-context learning. A reward is awarded only if a task is completed successfully.

**Setup.** We evaluate the model performance on four different tasks: *Pick & Place Objects, Stack Objects, Shape Sorter Placement, and Table Wiping.* To comprehensively assess instruction-following capabilities, the benchmark categorizes task instructions into five groups: (i) Base, which requires fundamental task-solving skills, (ii) Common Sense, which requires general world knowledge to resolve indirect objective references, (iii) Complex instructions, which involves longer, possibly distracting context that obscures the primary command, (iv) Spatial reasoning, where objects are referred to by their spatial relationships to other objects, and (v) Visual understanding, which requires recognizing objects based on visual attributes such as color or shape. All models are evaluated with a sampling temperature of 0. For comparison, the experimental results of GPT-4o-mini [18] and Gemini-1.5-flash [62] are adopted directly from the values reported in their paper.

**Result.** In Table 4, the model trained with the ROBOT-R1 framework achieves an approximate 31% performance increase over the baseline Qwen2.5-7B-Instruct model. In particular, in the Base category, which utilizes instructions similar to those used during ROBOT-R1 training, the model achieves nearly double the performance (6.3% → 12.5%). This indicates that the embodied reasoning capabilities learned via ROBOT-R1 transfer effectively to downstream tasks, even in setups that differ significantly from the training distribution. In contrast, the models fine-tuned with SFT failed to complete any tasks, highlighting the limitations of SFT approaches.

## 4.4 Ablation Study and Analysis

**Spatial reasoning ability.** To further evaluate whether enhanced spatial reasoning abilities of ROBOT-R1 – as shown in Section 4.2 and acquired through RL for optimizing reasoning paths in action prediction – generalize to broader spatial understanding scenarios, we evaluate the model's performance on the SpatialRGPT benchmark [20]. This benchmark is designed to evaluate 3D spatial comprehension capabilities in LVLMs. SpatialRGPT consists of two main components: *a qualitative evaluation*, which tests understanding of relative positional relationships between objects in an image, and *a quantitative evaluation*, which requires accurate prediction of numerical spatial values. As shown in Table 5, the model trained with ROBOT-R1 achieves substantial performance gains, with an approximate improvement of 40% in qualitative tasks and around 60% improvement in quantitative tasks. These improvements are particularly notable when compared to the performance of models trained solely with SFT, highlighting the effectiveness of RL-optimized reasoning in developing transferrable spatial understanding.

**Validation of the ROBOT-R1 bench.** To confirm the validity of the ROBOT-R1 Bench's LLM-as-judge, we measure the Pearson correlation between LLM and human assigned score. Considering

Table 5: **SpatialRGPT-Bench results.** Accuracy (%) of ROBOT-R1, SFT baselines in evaluating the performance of VQA tasks from SpatialRGPT-Bench [20], which measures (a) how LVLMs accurately predict the positional relations between objects, and (b) how LVLMs precisely estimate the absolute object positions, where the prediction is considered correct if it lies within ±25% of the ground-truth values. In contrast to SFT models that struggle to generalize to novel spatial reasoning tasks, ROBOT-R1 effectively transfers to these spatial reasoning tasks.

| (a) **Quantitative results** | | | | | | | |
|---|---|---|---|---|---|---|---|
| Model | Direct Dist. | Horiz. Dist. | Vert. Dist. | Width | Height | Direction | Quant. Avg |
| Qwen2.5-VL-7B-Ins [56] | 1.35 | 9.02 | 8.49 | 6.77 | 9.77 | 28.04 | 9.88 |
| w Direct SFT | 4.05 | 3.28 | 0.94 | 0.75 | 4.51 | **42.06** | 8.41 |
| w CoT SFT | 1.35 | 2.46 | 0 | 3.01 | 6.77 | 38.32 | 7.88 |
| w **ROBOT-R1(Ours)** | **6.08** | **14.75** | **22.64** | **21.8** | **19.55** | 12.15 | **15.89** |

| (b) **Qualitative results** | | | | | | | |
|---|---|---|---|---|---|---|---|
| Model | Below/Above | Left/Right | Big/Small | Tall/Short | Wide/Thin | Behind/Front | Qual. Avg |
| Qwen2.5-VL-7B-Ins [56] | 35 | **50.48** | 6.6 | 17.86 | 27.88 | 36.36 | 29.07 |
| w Direct SFT | 3.33 | 17.14 | 0 | 9.82 | 0 | 0 | 5.02 |
| w CoT SFT | 15.83 | 33.33 | **45.28** | **39.28** | **47.12** | 35.45 | 35.62 |
| w **ROBOT-R1(Ours)** | **42.5** | 41.9 | 31.13 | 37.5 | 38.46 | **52.72** | **40.79** |

Table 6: **Pearson correlation** between human and GPT-4o [18] judgments on ROBOT-R1 Bench. Computed over 80 responses (from ROBOT-R1 and GPT-4o), where human scores are defined as the median of 8 human expert annotations.

| | Planning | High-level Action | Movement | Spatial |
|---|---|---|---|---|
| Pearson Correlation | 0.3315 | 0.8974 | 0.8931 | 0.8961 |

the cost for collecting human annotations, we sample 40 problems from the ROBOT-R1 Bench and collect judgments from 8 participants with expertise in robotics. The experiment is conducted as a blind evaluation of responses produced by ROBOT-R1 and GPT-4o on a total of 80 problems. We define the human score as the median across the 8 participants. As shown in Table 6, the Pearson correlations between LLM and human scores are close to 0.9 for all tasks except planning, supporting the reliability of ROBOT-R1 Bench. For the planning task, we observe a lower correlation, likely because its longer and more open-ended responses allow multiple valid solutions, making precise automatic judging inherently more challenging. Notably, the tasks with higher correlations, such as high-level action, movement and spatial, are also where ROBOT-R1 achieves the largest performance gains, further supporting the validity of the improvements achieved by ROBOT-R1.

**Effect of different RL algorithms.** We evaluate the effectiveness of GRPO used to train ROBOT-R1 by training additional models with RLOO [64] and REINFORCE++ [65], two RL algorithms commonly employed for LLM optimization. We use the default hyperparameters provided by the workspace [4] and evaluate all models on the Robot-R1 Bench. Table 7 shows that RLOO achieves performance comparable to GRPO; however, REINFORCE++ performs worse on

Table 7: **ROBOT-R1 Bench results** on different RL algorithms.

| Model | Planning | High-level Action | Movement | Spatial |
|---|---|---|---|---|
| GRPO [46] | 1.44 | 1.30 | 0.76 | 1.51 |
| RLOO [64] | 1.56 | 1.54 | 0.68 | 1.52 |
| REINFORCE++ [65] | 1.40 | 1.08 | 0.62 | 1.38 |

average across tasks. We hypothesize that this is due to weaker reward normalization. Specifically, REINFORCE++ performs batch-level reward normalization, which often induces higher variance than the prompt-query-level normalization applied in GRPO and RLOO, leading to less stable learning and limited performance improvements. These findings indicate that RL algorithms that incorporate variance-reducing mechanisms, such as GRPO or RLOO, are beneficial for ROBOT-R1.

**Robustness across random seeds.** To evaluate whether ROBOT-R1 consistently yields performance gains across different random seeds, we conduct two additional experimental runs on the same dataset while changing only the random seed for online-sampling. We then evaluate all models on the Robot-R1 Bench. As shown in Table 8, ROBOT-R1 consistently achieves higher performance than

---

[4] https://github.com/hiyouga/EasyR1

Table 8: **Robot-R1 Bench results across random seeds.** ROBOT-R1 consistently improves performance over the non-finetuned original model (Qwen2.5-VL-7B-Ins [56]) across different random seeds, showing strong robustness to seed variability.

| Model | Planning | High-level Action | Movement | Spatial |
|---|---|---|---|---|
| Qwen2.5-VL-7B-Ins [56] | 1.66 | 1.04 | 0.58 | 1.40 |
| w **ROBOT-R1(Ours)** 1st Seed | 1.44 | 1.30 | 0.76 | 1.51 |
| w **ROBOT-R1(Ours)** 2nd Seed | 1.70 | 1.46 | 0.64 | 1.66 |
| w **ROBOT-R1(Ours)** 3rd Seed | 1.56 | 1.50 | 0.54 | 1.60 |
| Seed Avg. | 1.57 | 1.42 | 0.65 | 1.59 |
| Seed Std. | 0.13 | 0.11 | 0.11 | 0.08 |

Table 9: **Ablation study.** Evaluation results on the ROBOT-R1 Bench comparing different dataset designs, varying auxiliary tasks, i.e., waypoint prediction (WP), current state prediction (SP), and movement prediction (MP), and the question format.

| QA Type | | Task Type | | | Low Level Control | | | High Level Control | | |
|---|---|---|---|---|---|---|---|---|---|---|
| Open-End | MCQA | WP | SP | MP | Movement | Spatial | Avg | Planning | High-level Action | Avg |
| ✓ | - | ✓ | ✗ | ✗ | 0.68 | 1.37 | 1.03 | **1.60** | 1.26 | **1.43** |
| ✓ | - | ✓ | ✓ | ✗ | 0.48 | 1.03 | 0.76 | 1.48 | 1.16 | 1.17 |
| - | ✓ | ✓ | ✗ | ✗ | 0.40 | 1.42 | 0.91 | 1.44 | 1.22 | 1.33 |
| - | ✓ | ✓ | ✓ | ✗ | 0.54 | 1.51 | 1.03 | 1.50 | 1.24 | 1.37 |
| - | ✓ | ✓ | ✓ | ✓ | **0.76** | **1.51** | **1.14** | 1.44 | **1.30** | 1.37 |

non-finetuned model across all seeds. The improvements remain significant when accounting for variance. These results demonstrate that training with ROBOT-R1 is robust to seed variability.

**Impact of QA design on performance.** Table 9 demonstrates the effects of different QA design choices on model performance. Our results show that adding auxiliary tasks progressively improves overall performance. We also evaluate an alternative approach that uses open-ended generation instead of multiple-choice question answering (MCQA). In this setup, the model directly generats next waypoint values in an open-ended format and uses $\text{clip}(1 - \text{L1}, 0, 1)$ as the reward function, where the L1 represents the distance between the predicted and ground-truth states. While the open-ended approach initially outperforms single-task MCQA, it exhibits significant performance degradation when auxiliary tasks – such as current-state prediction – are introduced. Consequently, the MCQA framework demonstrates superior results. These findings highlight the critical role of question structure design in reinforcement learning for embodied reasoning.

**Qualitative analysis.** We find that the evolution of reasoning patterns during ROBOT-R1 training diverges notably from trends reported in Deepseek-R1 [17]. While reasoning models for mathematical and coding tasks tend to develop longer reasoning chains over time, ROBOT-R1 exhibits the opposite pattern of producing progressively shorter and more focused reasoning (see Appendix D for detailed examples). Qualitatively, the model initially generates broad planning or high-level action reasoning in an overview format. Over time, these are gradually replaced by more concise, logically structured reasoning traces that directly link relevant embodied reasoning components.

## 5 Conclusion

We have proposed ROBOT-R1, a novel training framework that enhances the embodied reasoning capabilities of Large Vision-Language Models in robotic domains. Our approach trains models to predict the next state based on image observations and current states from expert demonstrations, using explicit reasoning processed optimized through reinforcement learning. Our extensive experiments demonstrate that models trained with ROBOT-R1 not only outperform Supervised Fine-Tuning (SFT) based approaches on embodied reasoning tasks but also exhibit consistent performance improvements across diverse embodied tasks while SFT methods often suffer from performance degradation.

## Acknowledgements

This work was supported by Institute for Information & communications Technology Promotion(IITP) grant funded by the Korea government(MSIT) (No.RS-2019-II190075 Artificial Intelligence Graduate School Program(KAIST); No. RS-2024-00509279, Global AI Frontier Lab) and RLWRLD, Inc.

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

# A    Limitation and Social Impact

**Limitation.** In this paper, we only considered the xyz positions as state, without considering the rotation of the end-effector and the manipulation of the gripper. Therefore, ROBOT-R1 is hard to achieve reasoning about rotation or gripper movement, which are essential for complex tasks. Integrating these additional state dimensions would enhance ROBOT-R1's embodied reasoning ability for complex tasks.

**Social impact.** ROBOT-R1 proposes a method that enhances embodied reasoning for robot control using small LVLMs (under 10B parameters) with limited data. This approach is easily accessible to research labs and small companies working on robotics. We expect ROBOT-R1 to accelerate robotics research, facilitating faster application and adoption of robotic technologies in society. However, there is a potential risk that robots trained by reinforcement learning execute unintended actions to achieve their goals. This potentially leads to diminished human oversight, hence the design of complementary reward to mitigate such risk should be also explored in the future.

# B    Experiment Details

## B.1    Computing Cost

All experiments are conducted on a single node consisting of four A100 80GB GPUs. Training ROBOT-R1 task approximately 12 hours for 5 epoch using the 7.5K prompts.

## B.2    Prompt Template for Generating Training Dataset

In this section, we provide the prompt templates used to generate the training datasets. To generate the Multiple-Choice Question Answering (MCQA) dataset as denoted in Section 3.2, we utilize three different prompt templates: the waypoint prediction MCQA prompt template (Figure 4), the state prediction MCQA prompt template (Figure 5), and the movement prediction MCQA prompt template (Figure 6). These templates take the current state, task instructions, and question about problem.

For generating the SFT dataset, we employ a prompt structure analogous to the MCQA format. Specifically, we use two different prompt templates for both Direct SFT and CoT SFT: the waypoint prediction SFT prompt template (Figure 7) and the current state prediction SFT prompt template (Figure 8). However, the SFT dataset, unlike the ROBOT-R1 dataset, includes not only prompts but also answers. The Direct SFT dataset is structured to generate immediate answers, while the CoT SFT dataset incorporates a reasoning path for the waypoint prediction task. The CoT SFT dataset is designed to perform planning, high-level action reasoning, and movement prediction sequentially before predicting the final answer. The planning and high-level action reasoning components were defined using human-created scripts.

**Waypoint prediction MCQA**

# You are Franka Robot Assistant: Task Planning and Execution System

## Task description
{task_description}

## Visual Input

You will receive a single combined image for scene understanding:
- <image>: front view of the workspace

## Coordinate System

The world coordinate frame follows these conventions:
- This is based on the front view. (Wrist view has the Y-axis (left and right) opposite)
- X-axis: Front of table (positive) to back of table (negative)
- Y-axis: Left (negative) to right (positive)
- Z-axis: Down toward floor (negative) to up toward ceiling (positive)
- World origin [0.25, 0, 0.752] is at the center of the table surface

## Robot Specifications
- Gripper dimensions: 0.06m width (x-direction) × 0.2 length (y-direction) × 0.09 height (z-direction), with fingers 0.04 in length

## Current Robot State
Position: [{x}, {y}, {z}]

### Choice Question
Based on the provided image and current robot state, predict the next waypoint position [x, y, z]
Choose the most accurate option:

[[A]] {A}
[[B]] {B}
[[C]] {C}
[[D]] {D}

## Output Format

You FIRST think about the reasoning process as an internal monologue and then provide the final answer.
The reasoning process MUST BE enclosed within <think> </think> tags.
The final answer MUST BE enclosed within <answer> </answer> tags.

Example output format:

<think>
[detailed reasoning process]
</think>
<answer>
[[A]]
</answer>

Figure 4: **Waypoint prediction MCQA prompt template**

**Current State prediction MCQA**

# You are Franka Robot Assistant: Task Planning and Execution System

## Robot Task description
{task_description}

## Visual Input

You will receive a single combined image for scene understanding:
- <image>: front view of the workspace

## Coordinate System

The world coordinate frame follows these conventions:
- This is based on the front view. (Wrist view has the Y-axis (left and right) opposite)
- X-axis: Front of table (positive) to back of table (negative)
- Y-axis: Left (negative) to right (positive)
- Z-axis: Down toward floor (negative) to up toward ceiling (positive)
- World origin [0.25, 0, 0.752] is at the center of the table surface

## Robot Specifications
- Gripper dimensions: 0.06m width (x-direction) × 0.2 length (y-direction) × 0.09 height (z-direction), with fingers 0.04 in length

### Choice Question
Let's predict the current robot state base on image

[[A]] {A}
[[B]] {B}
[[C]] {C}
[[D]] {D}

## Output Format

You FIRST think about the reasoning process as an internal monologue and then provide the final answer.
The reasoning process MUST BE enclosed within <think> </think> tags.
The final answer MUST BE enclosed within <answer> </answer> tags.

Example output format:

<think>
[detailed reasoning process]
</think>
<answer>
[[A]]
</answer>

Figure 5: **Current state prediction MCQA prompt template**

> **Movement prediction MCQA**
>
> # You are Franka Robot Assistant: Task Planning and Execution System
>
> ## Task description
> {task_description}
>
> ## Visual Input
>
> You will receive a single combined image for scene understanding:
> - <image>: front view of the workspace
>
> ## Coordinate System
>
> The world coordinate frame follows these conventions:
> - This is based on the front view.
> - X-axis: Front of table (positive) to back of table (negative)
> - Y-axis: Left (negative) to right (positive)
> - Z-axis: Down toward floor (negative) to up toward ceiling (positive)
>
> ### Choice Question
> What movements are needed to get to the next keypoint to perform the task?
>
> [[A]] {A}
> [[B]] {B}
> [[C]] {C}
> [[D]] {D}
>
> ## Output Format
>
> You FIRST think about the reasoning process as an internal monologue and then provide the final answer.
> The reasoning process MUST BE enclosed within <think> </think> tags.
> The final answer MUST BE enclosed within <answer> </answer> tags.
>
> Example output format:
>
> <think>
> [detailed reasoning process]
> </think>
> <answer>
> [[A]]
> </answer>

Figure 6: **Movement prediction MCQA prompt template**

---

**Waypoint prediction QA for SFT**

**Prompt**

# You are Franka Robot Assistant: Task Planning and Execution System

## Task description
{task_description}

## Visual Input

You will receive a single combined image for scene understanding:
- **<image>**: front view of the workspace

## Coordinate System

The world coordinate frame follows these conventions:
- This is based on the front view. (Wrist view has the Y-axis (left and right) opposite)
- X-axis: Front of table (positive) to back of table (negative)
- Y-axis: Left (negative) to right (positive)
- Z-axis: Down toward floor (negative) to up toward ceiling (positive)
- World origin [0.25, 0, 0.752] is at the center of the table surface

## Robot Specifications
- Gripper dimensions: 0.06m width (x-direction) × 0.2 length (y-direction) × 0.09 height (z-direction), with fingers 0.04 in length

## Current Robot State
Position: [Current State]

Let's determine the next robot state to execute task

---

**Answer for CoT SFT**

Plan: 1. Move the gripper to align vertically and position just above the red button. 2. Move the gripper down for pressing the button.
Subtask: Move the gripper to align vertically and position just above the red button.
Move: slightly move front, slightly move left, and move down.
Answer: [Answer]

---

**Answer for Direct SFT**

[Answer]

---

Figure 7: **Waypoint prediction QA template for SFT**

```
┌──────────────────────────────────────────────────────────────────────────┐
│ Current State prediction QA for SFT                                        │
├──────────────────────────────────────────────────────────────────────────┤
│                                                                            │
│                               Prompt                                       │
│                                                                            │
│  # You are Franka Robot Assistant: Task Planning and Execution System      │
│                                                                            │
│  ## Task description                                                       │
│  {task_description}                                                        │
│                                                                            │
│  ## Visual Input                                                           │
│                                                                            │
│  You will receive a single combined image for scene understanding:         │
│  - <image>: front view of the workspace                                    │
│                                                                            │
│  ## Coordinate System                                                      │
│                                                                            │
│   The world coordinate frame follows these conventions:                    │
│  - This is based on the front view. (Wrist view has the Y-axis (left and   │
│    right) opposite)                                                         │
│  - X-axis: Front of table (positive) to back of table (negative)           │
│  - Y-axis: Left (negative) to right (positive)                             │
│  - Z-axis: Down toward floor (negative) to up toward ceiling (positive)    │
│  - World origin [0.25, 0, 0.752] is at the center of the table surface     │
│                                                                            │
│  ## Robot Specifications                                                   │
│  - Gripper dimensions: 0.06m width (x-direction) × 0.2 length (y-direction)│
│  × 0.09 height (z-direction), with fingers 0.04 in length                  │
│                                                                            │
│  Let's predict the current robot state base on image                       │
│  ────────────────────────────────────────────────────────────────────     │
│                               Answer                                       │
│                                                                            │
│  [Answer]                                                                  │
└──────────────────────────────────────────────────────────────────────────┘
```

Figure 8: **Current state prediction QA template for SFT**

## B.3 Benchmark Setup

To reduce variance in all benchmarks, we set the temperature of the model being evaluated to 0. Meanwhile, the ROBOT-R1 Bench uses GPT-4o [18] for LLM-as-judge, and we set the temperature of GPT-4o to 1 (default temperature) for judgment tasks.

**ROBOT-R1 Bench details.** The evaluation dataset consists of (1) images corresponding to observations, (2) questions related to embodied reasoning, and (3) human-written ground truth answers. The evaluated models receive the images and questions as input to generate responses. To improve the accuracy of answer generation, each model receives the following system prompt for every sampling process (see Figure 9). The evaluator, GPT-4o, provides scores between 0 and 3 points using the LLM-as-judge template prompt (see Figure 10) with the question, the response of the evaluated model, the predefined ground truth answer and the evaluation rubric. To sum up, full evaluation process is conducted as shown in Table 10.

## B.4 Benchmark Setup

To reduce variance in all benchmarks, we set the temperature of the model being evaluated to 0. Meanwhile, the ROBOT-R1 Bench uses GPT-4o [18] for LLM-as-judge, and we set the temperature of GPT-4o to 1 (default temperature) for judgment tasks.

**ROBOT-R1 Bench details.** The evaluation dataset consists of (1) images corresponding to observations, (2) questions related to embodied reasoning, and (3) human-written ground truth answers. The evaluated models receive the images and questions as input to generate responses. To improve the

Table 10: **Example of the working process of ROBOT-R1 Bench**

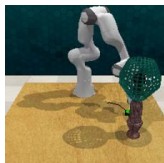

**Question:** What is the complete sequence of high-level actions needed to turn on the lamp?

**Model Raw Answer:** `<think>`To determine the sequence of actions needed to turn on the lamp, we need to consider the typical steps required for such a task. The lamp appears to be a standalone object with no visible control panel or buttons in the image provided. Assuming the lamp can be turned on by a remote control or a switch, the following steps would be logical: 1. Identify the lamp's location relative to the robot. 2. Determine if there is a remote control or switch accessible to the robot. 3. If a remote control or switch is present, use the robot's arm to interact with it. 4. If the lamp is directly accessible, move the robot's arm close to the lamp and interact with it. Since the image does not show a remote control or switch, and the lamp itself is not directly accessible, the robot might need to perform a series of actions to reach the lamp, such as moving towards it, interacting with it, and possibly using a tool to activate it.`</think>` `<answer>`1. Move the robot arm towards the lamp. 2. Interact with the lamp to turn it on.`</answer>`

**Model Answer:** 1. Move the robot arm towards the lamp. 2. Interact with the lamp to turn it on.

**Ground Truth:** 1. Position the robot's gripper near the lamp switch. 2. Press the lamp switch.

**LLM-as-Judge Result:** The model's answer correctly captures the intent and sequence of actions (approaching and turning on the lamp), but it uses more general terms ("move arm towards the lamp" and "interact with the lamp to turn it on") rather than explicitly stating positioning near the lamp switch and pressing it. While the core meaning is mostly preserved, the model omits the specific details of positioning near the switch and pressing it, making it slightly less precise.

**Score:** 2

Table 11: **Robot-R1 Bridge Bench results**

| Model | Planning | High-level Action | Movement | Spatial |
|---|---|---|---|---|
| Gemini-1.5 Pro [62] | 1.70 | 1.16 | 0.39 | 1.65 |
| Qwen2.5-VL-7B-Ins [56] | 1.20 | 0.89 | 0.46 | 1.53 |
| w **ROBOT-R1(Ours)** | 1.38 | 1.10 | 0.45 | 1.56 |

accuracy of answer generation, each model receives the following system prompt for every sampling process (see Figure 9). The evaluator, GPT-4o, provides scores between 0 and 3 points using the LLM-as-judge template prompt (see Figure 10) with the question, the response of the evaluated model, the predefined ground truth answer and the evaluation rubric. To sum up, full evaluation process is conducted as shown in Table 10.

## C   Additional Evaluation and Analysis

In this section, we present additional evaluations and analyses to further validate the effectiveness of ROBOT-R1. For efficiency, all experiments in this section are trained for one epoch before evaluation.

**Robot-R1 Bench on real robot.** To further investigate whether the embodied reasoning capability learned by ROBOT-R1 can transfer to real-world robot settings, we extend the ROBOT-R1 Bench to the ROBOT-R1 Bridge Bench [66]. The problem construction and evaluation procedure remain identical to the original benchmark, but we use 100 randomly sampled images from real-robot demonstration videos in BridgeV2 [66]. Table 11 shows the evaluation results on the ROBOT-R1 Bridge Bench. Despite being trained only on RLBench simulation images, the model trained with ROBOT-R1 demonstrates notable performance gains, indicating that the high-level embodied

Table 12: **VLABench LVLM Bench results**

| Model | M&T | Common-Sense | Semantic | Spatial | Physical-Law | Complex |
|---|---|---|---|---|---|---|
| Qwen2.5-VL-7B-Ins [56] | 39.02 | 40.78 | 37.64 | 35.83 | 39.20 | 38.51 |
| w **ROBOT-R1(Ours)** | 46.58 | 41.5 | 37.62 | 40.33 | 25.33 | 33.87 |
| w **CoT-SFT** | 0 | 0 | 0 | 0 | 0 | 0 |

Table 13: **Results of the cold start experiment on ROBOT-R1 Bench**

| Model | Planning | High-level Action | Movement | Spatial |
|---|---|---|---|---|
| Qwen2.5-VL-7B-Ins [56] | 1.66 | 1.04 | 0.58 | 1.40 |
| + **High Quality CoT-SFT** | 0.98 | 1.34 | 0.88 | 1.4 |
| + **ROBOT-R1(Ours)** | 1.22 | 1.44 | 0.88 | 1.43 |

reasoning ability acquired through ROBOT-R1 successfully transfers to real-robot environments. However, improvements in the embodied reasoning tailored for low-level action are relatively limited, which we attribute to differences in camera viewpoints and coordinate systems between RLBench and BridgeV2.

**Evaluation on VLABench.** We further evaluate whether the model trained with ROBOT-R1 generalizes to other robot agent environments beyond RLBench, using the VLABench [67] evaluation pipeline under the CoT setup. As shown in Table 12, the performance improves on the *Mesh & Texture (M&T)* and *Spatial* tasks, while it decreases slightly on the *Physical Law* and *Complex Reasoning* tasks. We hypothesize that this performance drop stems from the fundamental gap between these task types and the abilities learned via ROBOT-R1. For instance, *Physical Law* tasks include questions such as "We have three objects with different densities. Choose the object with the smallest density," while *Complex Reasoning* tasks include questions such as "Please rearrange these books in chronological order starting from the first published to the latest." These tasks primarily require reasoning unrelated to embodied manipulation. In contrast, the *M&T* and *Spatial* tasks involve instructions such as "Add salt to the dish" or "Add the second condiment from the bottom to the dish," which demand spatial understanding and movement prediction directly targeted by ROBOT-R1. These findings demonstrate that the embodied reasoning capabilities learned through ROBOT-R1 can be effectively transferred to new robotic environments beyond RLBench.

**Cold-start strategy.** Training with high-quality SFT reasoning data followed by RL is a promising direction, as demonstrated in Guo et al. [17]. To evaluate whether ROBOT-R1 remains effective in such a cold-start scenario, we conducted experiments using GPT-4.1-mini [68] to generate high-quality CoT-SFT data and then applied ROBOT-R1 fine-tuning on top of the SFT trained model with high-quality CoT-SFT data. Specifically, the SFT data are generated by conditioning on human-annotated ground-truth planning and high-level action information, and prompting the model to produce CoT-style responses under the MCQA prompt used for RL. All SFT training settings followed those in the main experiment. As shown in Table 13, ROBOT-R1 demonstrates consistent performance improvements even on top of the SFT model, indicating the effectiveness on cold-start settings. 'However, SFT-only training still shows decreased planning task performance and limited improvement in spatial tasks, implying that even with high-quality supervision, models remain vulnerable to forgetting and poor generalization.

# D   More Quantitative Results

In this section, we present quantitative metrics observed during the ROBOT-R1 training process. Specifically, we examine how the reward and response length change during training. Additionally, we include results for the Open-End approach trained ROBOT-R1 in Section 4.4. Open-End approach, unlike MCQA approach, directly generats next waypoint values and use L1 distance for reward which is define between the predicted and ground-truth states. Figure 11a illustrates how the MCQA accuracy and the accuracy reward obtained from 1-L1 distance change during the training process.

Table 14: **VLA performance on LIBERO**

| Model | Long | Goal | Objective | Spatial | Avg. |
|---|---|---|---|---|---|
| Qwen2.5-VL-7B-Ins [56] | 44.0 | 61.6 | 83.4 | 86.8 | 69.0 |
| w **ROBOT-R1(Ours)** | 45.8 | 73.8 | 80.4 | 87.0 | 71.8 |

Table 15: **VLA performance on real robot**

| Model | PnP 1 | PnP 2 | PnP 3 | PnP 4 | Avg. |
|---|---|---|---|---|---|
| Qwen2.5-VL-7B-Ins [56] | 25 | 0 | 20.83 | 20.83 | 16.67 |
| w **ROBOT-R1(Ours)** | 25 | 29.17 | 25 | 16.67 | 23.96 |

These results show that GRPO training is being performed effectively. As shown in Figure 11b, ROBOT-R1 also demonstrates that the response character length decreases throughout the training process. We hypothesize that this occurs as the reasoning process transitions from a summary format to a narrative format, during which redundant or unnecessary information is removed from the responses. Detailed analysis is provided in Section E.

# E   Qualitative Results

This section presents the differences between model responses with ROBOT-R1 and Qwen2.5-VL-7b-ins (ROBOT-R1's base model). Figure 12 shows response examples of ROBOT-R1 trained with the MCQA approach. Figure 13 shows response examples of ROBOT-R1 trained with the Open-End approach from Section 4.4. Commonly, Qwen2.5-VL-7b-Ins demonstrates weak connections between reasoning process components and tends to generate responses in a summary format rather than a narrative format. However, after ROBOT-R1 training, the reasoning processes become coherently connected and shift toward a narrative format in their responses.

# F   Experiments on Vision-Language-Action Models

Finally, we evaluate whether training with ROBOT-R1 not only improves embodied reasoning but also enhances downstream robot control performance. We utilize the GR00T-N1.5 [69] repository for attaching a new diffusion policy head top on the LVLM backbone. Experiments are conducted in both the LIBERO simulation [70] environment and on a real robot.

**LIBERO simulation.** In LIBERO simulation, the environment is simulated using a Franka Panda Arm and includes four evaluation categories, such as, Spatial, Object, Goal, and Long. Each category consists of 10 tasks, and every task is evaluated over 50 rollouts. The action policy is trained using all given fine-tuning data with a batch size of 32 for 60k steps, following the original diffusion policy training setup while keeping all other hyperparameters identical to the GR00T-N1.5 repository. Table 14 presents the success rates in the four LIBERO categories. The model trained with ROBOT-R1 outperforms the baseline on average. In particular, the Goal category shows a remarkable performance improvement, which we attribute to its complex manipulation requirements that are closely aligned with the reasoning enhancements encouraged by ROBOT-R1. Although the Objective category shows a slight performance drop, we attribute this to the limited object diversity in RLBench during ROBOT-R1's training stage.

**Real robot experiment.** In real robot experience, we utilize a Franka Research 3 arm to perform four different pick-and-place tasks, each involving four distinct objects: a teddy bear, box, cup, and sponge. Each task is evaluated 24 times, resulting in 96 total trials to calculate the success rate. For training, each object has 60 demonstration trajectories, leading to 240 training examples per task. For each task, we train a separate action diffusion policy using its corresponding demonstrations for 30k steps with a batch size of 32, while keeping all other hyperparameters consistent with the simulation setup. Table 15 shows that training the action diffusion policy with ROBOT-R1 improves the average success rate from 16.67% to 23.96%, confirming ROBOT-R1 effectiveness in enhancing real-world robot control performance.

## System Prompt for Sampling in ROBOT-R1 Bench

**System Prompt Format**

You are an AI that accurately answers questions about robot actions and spatial relationships.
Follow these rules strictly:
1. Answer ONLY what is asked in the question.
2. Do not include any purpose or objective of actions (remove all 'to...' phrases).
3. Do not include any additional descriptive information.
4. Keep answers concise and focused on the core information.
5. Remove all unnecessary details about the current state or conditions.
6. Direction should be judged based on the viewpoint in the image.
* up: away from the ground
* down: toward the ground
* forward: toward the camera (where the image was taken from)
* backward: away from the camera
* right: to the right side from the camera's perspective
* left: to the left side from the camera's perspective
TASK GUIDELINES:
{task_description}

Additional Style Requirements:
- Use simple and clear English.
- Focus on semantic correctness, not stylistic variation.
- Keep sentences short and remove unnecessary details.
- If multiple directions are involved, combine them clearly (e.g., 'Move down and slightly right').

---

**Task Description Examples**

---

**Task Type: "spatial"**
- Focus only on relative positions and spatial relationships between objects.
- Do not describe any action or movement.
- Only describe the current spatial configuration.
- Example Answer: "The gripper is above the cup, offset to the right."

---

**Task Type: "planning"**
- List major actions in chronological order (1., 2., 3., ...).
- Each step should describe only the action, without mentioning the purpose.
- Example Answer: "1. Move the robot arm above the cup. 2. Lower the gripper to grasp the cup.
3. Lift the cup upward."

---

**Task Type: "high_level action"**
- Focus on the immediate next meaningful subtask (a single self-contained action).
- Describe WHAT needs to be done, not HOW to do it.
- Only the immediate next step, not the final goal.
- Example Answer: "Move the gripper closer to the button."

---

**Task Type: "movement"**
- Specify only mechanical movements and gripper state changes.
- Use the robot's perspective for directions:
- Describe only the very next physical movement.
- Example Answer: "Move down and slightly right."

Figure 9: **System prompt for sampling in ROBOT-R1 Bench**

## Evulation Rubric and LLM-as-Judge prompt

### Rubric

**0 points: Meaning completely different from ground truth**
- Answer has a completely different meaning from ground truth
- Key concepts and ideas are misinterpreted
- Contains information that contradicts ground truth

**1 point: Partially matches but with significant meaning differences**
- Some key points match but main meaning is different
- Contains significant misunderstandings of core concepts
- Has some correct information but overall meaning is incorrect

**2 points: Mostly matches in meaning but with minor differences**
- Main meaning and key points are correct
- Minor details or expressions are different
- Overall context and intent are preserved

**3 points: Meaning is equivalent to or more detailed than ground truth**
- Any expression that conveys the same basic meaning as ground truth
- Any description that leads to the same functional outcome
- Any expression that maintains the same spatial relationship between objects
- Any description that achieves the same goal through equivalent means
- Any expression that preserves the core meaning while using different words
- Any description that maintains the same context and intent
- Any expression that describes the same target location and action

**Additional Notes:**
1. Focus on semantic equivalence rather than exact wording
2. Different expressions are acceptable if they convey the same meaning
3. Consider the overall context and intent of the answer
4. Minor differences in expression are acceptable if the core meaning is maintained
5. Paraphrasing is considered a perfect match if the meaning is preserved
6. Different ways of describing the same action (e.g., 'press' vs 'flip' a switch) should be considered equivalent
7. The specific actor (robot vs human) should not affect the score if the action described is functionally equivalent
8. Different ways of expressing the same spatial relationship (e.g., 'over' vs 'toward') should be considered equivalent
9. Focus on whether the answer achieves the same functional outcome as the ground truth
10. Consider the answer as a 3 if it describes the same action or state using different but equivalent words

---

### System Prompt

You are an expert in evaluating the consistency between model's answer and ground truth answer. Please assign a score between 0-3 based on the given rubric and explain the reason in detail. You must respond in the format 'Score: [0-3] Reason: [explanation]' in a single line. You may use line breaks, but Score and Reason must be clearly separated and identifiable.

---

### User Prompt

Please evaluate how well the model's answer matches the ground truth answer.
Evaluation Criteria:
{rubric}
Question: {question}
Model's Answer: {model_answer}
Ground Truth Answer: {ground_truth}
Please assign a score between 0-3 and explain the reason in detail.

Figure 10: **Evaluation rubric and LLM-as-Judge prompt for model answer assessment.**

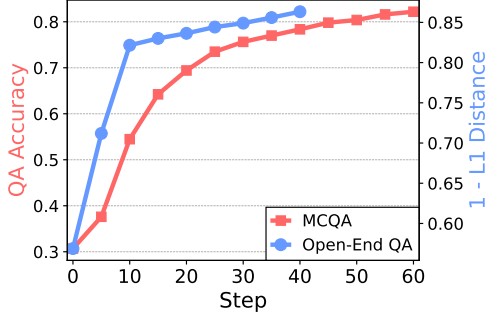
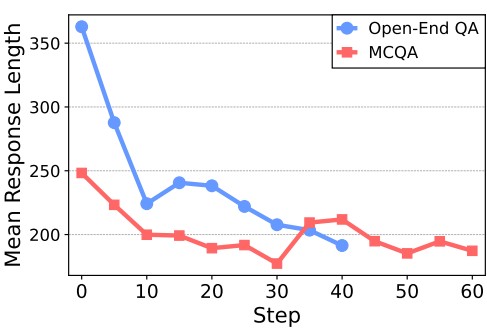

(a) Avg reward change during ROBOT-R1 training

(b) response length change during ROBOT-R1 training

Figure 11: **Training metrics for ROBOT-R1 across two learning approaches: Open-End QA and MCQA.** We conducted 10 epoch training for Open-End QA and 5 epoch training for MCQA (note that all models reported in the paper are based on 5 epoch training). Due to differences in dataset sizes between the two tasks, the graphs are plotted based on training steps. (a) Shows accuracy metrics for the MCQA task (left axis) and 1 - L1 distance for Open-End QA (right axis). (b) Shows mean response length over training steps for both task types.

**Question:**
You are Franka Robot Assistant: Task Planning and Execution System

Task description: lift the lid off the pan

[...]

Current Robot State: Position [0.265, -0.112, 1.154]

Choice Question: Based on the provided image and current robot state, predict the next waypoint position [x, y, z]. Choose the most accurate option:

[[A]] [0.072, 0.243, 0.996]
[[B]] [-0.07, 0.247, 0.937]
[[C]] [0.254, -0.197, 0.894]
[[D]] [0.187, -0.065, 1.144]

[...]

**GT Answer:** [[C]]

**Qwen2.5-VL-Ins Answer:**
<think>
To predict the next waypoint position for the robot to lift the lid off the pan, we need to consider the current position and the desired action. The robot is currently positioned above the pan, and the lid is on top of the pan. The goal is to lift the lid, which means the robot should move closer to the lid while maintaining a position that allows it to grasp the lid.

Let's analyze the options:

Option A: [0.072, 0.243, 0.996] - positioned above the lid, reasonable for grasping action
Option B: [-0.07, 0.247, 0.937] - further to the left, not efficient path
Option C: [0.254, -0.197, 0.894] - further back, not aligned with lid position
Option D: [0.187, -0.065, 1.144] - further up, not necessary for grasping

Option A has x and y coordinates close to the lid's position, and the z-coordinate is slightly lower than the current position, which is necessary for the robot to reach the lid. Therefore, the most accurate option is A.
</think>
<answer>
[[A]]
</answer>

reward: 0

**ROBOT-R1 Answer:**
<think>
To predict the next waypoint position for the robot to lift the lid off the pan, we need to consider the current position of the pan and the lid, as well as the robot's current position and orientation. The lid is currently on the pan, and the robot is positioned above the pan.

Given the current robot position [0.265, -0.112, 1.154], the pan is located close to the robot, and the lid is on top of the pan. We need to move the robot to a position where it can grasp the lid and lift it off the pan. The most accurate position to predict for the next waypoint would be one that allows the robot to reach the lid on the pan.

Looking at the coordinates of the pan and the lid, the most suitable position for the robot to reach the lid would be a position above the lid, but not directly on top of it, to ensure the robot can grasp the lid without hitting the pan. The option [[C]] [0.254, -0.197, 0.894] is the closest to the lid while still being above it, making it the most accurate choice for the next waypoint.
</think>
<answer>
[[C]]
</answer>

reward: 1

Figure 12: **Example of model response (MCQA)**

**Question:**
You are Franka Robot Assistant: Task Planning and Execution System

Task description: chuck way any rubbish on the table rubbish

[...]

Current Robot State: Position [0.275, -0.009, 1.448]

[...]

Let's determine the next robot state to execute task

---

**GT Answer:** [0.179, -0.026, 0.846]

---

**Qwen2.5-VL-Ins Answer:**
<think>
The task is to chuck away rubbish on the table. From the visual input, we can see that there are two red objects (possibly apples) and a crumpled piece of paper on the table. The black trash bin is present, indicating where the rubbish should be placed. The robot's gripper dimensions need to be considered to determine if it can pick up the paper. The gripper can easily pick up the paper, but the apples are larger and may require a different strategy or a second task. For this task, the focus is on the crumpled paper.

The robot's current position is [0.275, -0.009, 1.448]. The next step would be to move towards the crumpled paper, pick it up, and then move it to the trash bin.

1. Move the robot's arm towards the crumpled paper.
2. Open the gripper to pick up the crumpled paper.
3. Move the gripper to the trash bin.
4. Close the gripper and drop the paper into the trash bin.
5. Move the robot's arm back to its starting position.

Let's calculate the coordinates for the paper and the trash bin:
- The crumpled paper is located at approximately [0.2, -0.1, 0.75].
- The trash bin is located at approximately [0.4, -0.1, 0.75].

The robot will move from its current position to the paper and then to the trash bin.
</think>
<answer>
[0.2, -0.1, 0.75]
</answer>

---

L1 distance between target: 0.191

---

**ROBOT-R1 Answer:**
<think>
The task is to remove any rubbish from the table. Observing the image, there are two apples and a piece of paper on the table. The paper appears to be the "rubbish" as it is not a food item. The robot's gripper is positioned above the table, and the basket is nearby, suggesting it can be used to collect the rubbish. The robot needs to grab the paper and drop it into the basket.

The robot's current position is above the paper, and it needs to move closer to the paper to grab it. Once the paper is picked up, it should be moved to the basket and released.

Steps:
1. Move the robot's gripper closer to the paper.
2. Grasp the paper.
3. Move the paper towards the basket.
4. Release the paper in the basket.
</think>
<answer>
[0.275, -0.009, 0.85] # Move closer to the paper
</answer>

---

L1 distance between target: 0.117

---

Figure 13: **Example of model response (Open-End QA)**

