# OpenReview forum: "Robot-R1: Reinforcement Learning for Enhanced Embodied Reasoning in Robotics"
_NeurIPS.cc/2025/Conference — NeurIPS 2025 poster_

### Official Review · Reviewer_5ayi · 2025-06-13

**Clarity:** 2
**Significance:** 1
**Originality:** 2
**Rating:** 4
**Confidence:** 5

**Summary:**

This article presents Robot-R1, a framework designed to enhance embodied reasoning for robot control by leveraging reinforcement learning. Traditional approaches often rely on Supervised Fine-Tuning (SFT) using datasets that are heuristically constructed and not specifically optimized for improving robot control, leading to issues such as catastrophic forgetting and reduced generalization performance. Robot-R1 addresses these limitations by using DeepSeek-R1 learning approach to learn to predict the next keypoint state required for task completion, based on the current scene image and environment metadata derived from expert demonstrations. Experimental results demonstrate that models trained with Robot-R1 outperform SFT methods on embodied reasoning tasks.

**Questions:**

1. Since the training of Robot-R1 is conducted on MCQA constructed from expert demonstrations, (1) how is Robot-R1 adapted to Robot-R1 Bench? Or is it that Robot-R1 Bench is also in the form of MCQA, without the need for the model to generate open-ended answers? (2) How is Robot-R1 adapted to other benches such as EmbodiedBench Manipulation, which does not have answers in the form of MCQA?

2. As shown in Appendix B.2 Prompt Template for Generating Training Dataset, the answer formats for Direct SFT and CoT-SFT are open-ended, while the answer format for Robot-R1 compared in the main text of the paper is in the form of MCQA. I believe this comparison is unfair. Although the appendix contains a comparison of the two answer formats for Robot-R1 (multiple-choice and open-ended), there is no comparison between the open-ended answers of Robot-R1 and the baselines.

3. According to the results of EmbodiedBench Manipulation, GPT-4o achieves a 28.9% success rate in low-level manipulation, which is significantly better than Robot-R1. Should this result be included as one of the baselines in Table 4?

4. Appendix B.1 Computing Cost states, "Training the ROBOT-R1 task takes about 12 hours per epoch using 7.5K prompts." Regarding token costs, the phrase "using 7.5K prompts" is too vague. How long is a prompt? The total number of tokens consumed should be specified. Additionally, what about the cost of test? How many tokens are needed to test all tasks in the benchmark? And how many tokens do the baselines consume?

5. According to the results shown in Figure 11(a) in the appendix, the open-ended QA format of Robot-R1 outperforms the MCQA format of Robot-R1 (achieving higher QA accuracy with fewer training steps). So, why not use the open-ended format of Robot-R1 in the main text of paper? Moreover, why not train for one more epoch (thus reaching the same number of training steps as the MCQA format of Robot-R1) to verify whether the open-ended format can achieve higher asymptotic performance?

6. How can the accuracy of scoring using large models in the Robot-R1 Bench be ensured?

7. The abstract mentions that SFT data (1) is often constructed heuristically, not explicitly optimized for improving robot control; (2) leads to issues such as catastrophic forgetting and reduced generalization performance. However, Robot-R1 (or all papers using the R1 training paradigm) still uses dataset that is the same as or similar to SFT, which has the disadvantage of heuristic construction. Moreover, how can it be ensured that the R1 training paradigm does not suffer from catastrophic forgetting and reduced generalization performance? There seems to be no literature proving this point. The paper mentions it but does not provide proof.

**Ethical Concerns:**

["NO or VERY MINOR ethics concerns only"]

**Final Justification:**

- The author's response has addressed my concerns. And I will raise the score from 2 to 4.
- I still care a lot about the consistency of the terminology used in the paper, which indeed poses a significant barrier to reading. I hope the authors can make timely corrections.

**Limitations:**

yes

**Paper Formatting Concerns:**

There are no formatting issues in this paper.

**Quality:**

2

**Strengths And Weaknesses:**

**Strengths**

1. The overall approach presented in the paper is relatively straightforward and easy to understand.
2. Robot-R1 demonstrates improved performance compared to baseline methods.

**Weaknesses**

1. The core training methodology is derived from DeepSeek-R1, and the paper lacks further innovation in this aspect.
2. There are some confusing statements in line 182. For example, the distinction between "high-level action planning" and "subtask reasoning" is unclear. It appears that "high-level action" might be synonymous with "subtask," and similarly, "primitive movement reasoning" seems to overlap with "subtask reasoning." Additionally, the categorization of "primitive movement" as low-level reasoning is questionable, especially since it is unclear how primitive movements alone can control a robot. The terminology is inconsistent throughout the paper (e.g., switching between "high-level action planning" and "action planning," or "primitive movement reasoning" and "primitive action reasoning"), which makes it difficult to follow. It is recommended that the authors provide clear definitions and use consistent terminology. Visual aids, such as diagrams, could also help clarify these concepts.
3. There are formatting issues, such as Table 3 appearing before Table 2.
4. See Questions

---

> ### Author Rebuttal · Authors · 2025-07-31
>
> Dear Reviewer 5ayi,
>
> We sincerely appreciate your thoughtful comments. We have carefully considered each of your questions and provide detailed responses below.
>
> ---
>
> **[W1] Lack of novelty**
>
> We clarify that our contribution is NOT in applying the RL training method, GRPO from DeepSeek-R1, to a new domain, but in proposing a novel framework specifically tailored to address the unique challenges for enhancing the embodied reasoning of LVLM. Specifically, our contributions are following three folds:
>
> 1. Challenges in Applying RL Algorithms for Improving Embodied Reasoning of LVLM
> - Embodied QA lacks definitive answers, leading to suboptimal training targets.
> - Current AIs demonstrate poor embodied reasoning performance, so most annotations rely on human experts.
>
> 2. Core Innovation: Waypoint-based Embodied Reasoning
> - Since the main goal of robot control is action generation, we enhance overall embodied reasoning by predicting waypoints that are fixed targets and easy to achieve.
>
> 3. Methodological Advance: MCQA and Auxiliary Task Integration
> - Numeric states have limited representational power, making reasoning enhancement through state prediction alone challenging
> - We design diverse QA formats (MCQA, open-ended) and validate these approaches through experiments
> - To enhance performance, we further consider auxiliary training tasks such as state prediction to improve understanding of numeric states
>
> To summarize, robot-r1 suggests a novel embodied reasoning training framework that enhances performance using numeric state information without any human annotation. Our framework utilizes GRPO, but overcomes many practical challenges stated above while applying it to a robot domain.
>
> ---
>
> **[W2,W3] Terminology and format error**
>
> Thank you for your suggestion. We will clarify the concepts by providing clear definitions of the terms consistently. In particular, we will define these terms:
> - “planning” refers to the entire sequence of all high-level actions required to complete a task.
> - “high-level action reasoning” refers to linguistic high-level action expression
> - “movement reasoning” refers to a more accurate expression that indicates a concrete change in direction within the 3d space, e.g., “move up”, “move down”.
>
> And we will fix the table format in the final draft.
>
> ---
>
> **[Q1] Is Robot-R1 Bench MCQA-style? How does MCQA-trained Robot-R1 adapt to open-ended benchmarks?**
>
> All evaluation environments did not use the MCQA format, and all models were evaluated through open-ended answer generation. This is because Robot-R1 enhances overall embodied reasoning performance by solving MCQA through the reasoning process during training, i.e., the knowledge is easily generalizable to various domains.
>
> ---
>
> **[Q2] Unfair comparison between Robot-R1 and SFT**
>
> Our comparison is fair because we already did an open-ended generation evaluation for both Robot-R1 and SFT, as we also mentioned in our response to [Q1].
>
> ---
>
> **[Q3] GPT-4o performs better at EmbodiedBench Manipulation**
>
> We first clarify that GPT-4o can’t be a fair baseline, as it has a significantly larger model size than Qwen-2.5 VL 7B, which we used to train Robot-R1. As our goal is to develop the training methodology to enhance the embodied reasoning in robotics, our baseline is an initial backbone LLM (no fine-tuning) or other training methods such as SFT. The reason why we present the performance of GPT-4o and other LLMs is to provide a useful reference for better performance judgment.
>
> Additionally, EmbodiedBench is an agent benchmark, requiring not only embodied reasoning but also overall in-context learning capabilities. Meanwhile, Robot-r1 was trained to enhance embodied reasoning performance, not the performance of an LLM agent. Therefore, Robot-R1 is much unfavorable in this benchmark, compared to GPT-4o.
>
> ---
>
> **[Q4] Computing cost**
>
> We have disclosed all the prompt templates used in the experiments in B.2, allowing for easy token calculation. The average prompt length used in RL training is approximately 700 tokens, and that used in SFT is approximately 400 tokens.
>
> The benchmark also includes all prompts publicly. Each problem for evaluation consists of approximately 200 tokens, and the model takes approximately 20 minutes for one A100 to generate all responses for a total of 215 problems. The API cost for LLM-as-judge is approximately $2 for scoring all responses in the benchmark. Each LLM-as-Judge prompt consumes approximately 800 prompt tokens and 90 response tokens.
>
> ---
>
> **[Q5] Training Robot-R1 with open-ended QA format outperforms MCQA format at training accuracy**
>
> We clarify that the results from the open-ended QA format and MCQA format are not comparable in Figure 11(a), because the applied pivot of the y-axis is different between the MCQA and open-ended formats. The open-end QA’s y-axis indicates (1 - L1 loss) while MCQA’s y-axis is QA accuracy.
>
> ---
>
> **[Q6] How can the accuracy of scoring using large models in the Robot-R1 Bench be ensured?**
>
> We first note that prior works in the literature of LLM-as-judge have observed that the evaluation with strong LLMs such as GPT-4 and human evaluation exhibit high correlation [1,2,3].
>
> Nevertheless, to further investigate the validity of this approach for Robot-R1 Bench, we measured the correlation between real human scores and robot-r1 bench scores, by having human experts score responses in the same environment as the LLM-as-judge model scores responses in Robot-R1 Bench.
>
> Specifically, we conducted the test using a subset of 40 problems from the Robot-R1 bench with 8 participants. The test was conducted as a blind evaluation of Robot-R1 and GPT-4o responses, and we defined the median score obtained from this evaluation as the human score.
>
> \begin{array}{c|cccc}
> \hline
> & \textbf{Planning} & \textbf{Subtask} & \textbf{Movement} & \textbf{Spatial} \newline
> \hline
> \textbf{Pearson Correlation} & 0.3315 & 0.8974 & 0.8931 & 0.8961 \newline
> \hline
> \end{array}
>
> The following table shows the Pearson correlation between LLM and human scoring, which supports the reliability of the Robot-R1 bench. Meanwhile, planning shows a somewhat lower correlation coefficient. This is likely because planning tasks typically involve longer responses, which increases the possibility of multiple valid answers. This structural characteristic makes accurate evaluation inherently challenging for planning tasks.
>
> However, Robot-R1 particularly excelled in the other components, and we believe these results do not compromise the reliability of our experimental findings.
> We will further scale up the reliability assessment during the remaining period and include the results in the final draft.
>
> ---
>
> **[Q7] Why is RL better than SFT?**
>
> While the datasets used in RL training are constructed using specific rules, the model actually learns the embodied reasoning process that emerges throughout the training process, which is automatically generated and optimized.
>
> We also evaluated across diverse environments, including Robot-R1 bench, EmbodiedBench, and SpatialRGPT, which all involve different tasks from our training objective of waypoint prediction. We observed overall performance improvements compared to pre-trained models across these varied tasks. These results experimentally demonstrate the generalization capability of our RL approach.
>
> Meanwhile, research demonstrating that SFT lacks generalization and causes forgetting compared to reassigning incentivized RL methods is a widely known fact in LLM research and has been experimentally proven across a wide range of fields, math, coding, vision etc [4,5]. For example, RL is known to achieve strong generalization performance with minimal data compared with SFT, even with a single sample [6].
>
> ---
> [1] Zheng et al., Judging LLM-as-a-Judge with MT-Bench and Chatbot Arena, NeurIPS 2023 \
> [2] Liu et al., G-Eval: NLG Evaluation using GPT-4 with Better Human Alignment, EMNLP 2023 \
> [3] Dubois et al., Length-Controlled AlpacaEval: A Simple Way to Debias Automatic Evaluators, COLM 2024 \
> [4] Chu et al., SFT Memorizes, RL Generalizes: A Comparative Study of Foundation Model Post-training, ICML 2025 \
> [5] Guo et al. Deepseek-r1: Incentivizing reasoning capability in llms via reinforcement learning, Arxiv 2025 \
> [6] Wang et al., Reinforcement Learning for Reasoning in Large Language Models with One Training Example, Arxiv 2025
>
> ---
>
> If you have any further questions/concerns, please do not hesitate to let us know.
>
> Thank you very much,  \
> Authors

---

> > ### Comment · Reviewer_5ayi · 2025-08-01
> > **Reply to 'Rebuttal by Authors'**
> >
> > > Q1 & Q2.
> >
> > I now understand that the model evaluation is conducted in an open-ended format. However, I still have doubts about the fairness. This is because Robot-R1 was trained in the MCQA format, while SFT was trained in the open-ended format. If evaluation results of models trained in the MCQA format using SFT and SFT with CoT could be provided, the fairness would be more convincing.
> >
> > > Q7.
> >
> > Could the authors provide some intuitive examples to demonstrate that Robot-R1 has better generalization ability than SFT w. CoT (it would be even better if SFT w. CoT were trained on question - and - answer data in a format similar to the MCQA format used for Robot - R1), rather than simply explaining it through experimental metrics? For example, what embodied reasoning habits or processes were formed during the training of Robot-R1 that did not appear in the CoT data of SFT w. CoT but contribute to enhancing the model's reasoning and generalization abilities?

---

> ### Author Response · Authors · 2025-08-02
>
> **[q1 & q2] Unfair comparison between Robot-R1 and SFT**
>
> We first note that SFT has been conventionally trained in an open-ended QA format [1,2] and the majority of evaluations in this paper are conducted in an open-ended format as well. Hence, we believe this open-ended setup is valid and favorable for SFT.
>
> Nevertheless, to address your concern about fairness, we conducted two additional experiments using MCQA-SFT and MCQA-CoT-SFT approaches based on the datasets used for Robot-R1 training. The results are presented in the table below. SFT models trained with the MCQA formats still exhibit significantly lower performance compared to Robot-R1.
>
> Specifically, MCQA-CoT improved planning and spatial reasoning compared to open-ended CoT, but still underperformed even the non-fine tuned model and lost the previously observed primitive movement gains. We will include this result in the final draft.
>
>
> \begin{array}{c|cccc}
> \hline
> \text{Metric} & \text{planning} & \text{subask} & \text{movement} & \text{spatial} \newline \hline
> \text{Qwen} & 1.66 & 1.04 & 0.58 & 1.40 \newline \hline
> \text{Robot-R1} &1.44 & 1.30 & 0.76 & 1.51 \newline  \hline
> \text{MCQA-SFT} & 0 & 0.52 & 0.12 & 0.94 \newline
> \text{MCQA-CoT-SFT} & 1.22 & 1.28 & 0.58 & 0.37 \newline
> \text{Open-end-CoT-SFT} & 0.60 & 1.28 & 0.70 & 0.29 \newline  \hline
> \end{array}
>
>
> ---
>
>
> **[q7] Why is RL better than SFT?**
>
> We appreciate the reviewer’s insightful suggestion. Following your request, we provide intuitive examples to illustrate how Robot-R1 achieves better generalization than SFT w/ CoT. Specifically, we highlight two distinctive reasoning behaviors that emerged from reinforcement learning in Robot-R1, which are not observed in the SFT w/ CoT model:
>
> **1. Adaptive Reasoning Planning**
>   - [Robot-R1]: Through reinforcement learning, Robot-R1 learns to adaptively determine the type and score of reasoning needed to solve the current task. This adaptive planning enhances the model to generalize effectively to new tasks.
>   - [SFT w/ CoT Limitation]: SFT w/ CoT relies on fixed, template-based reasoning patterns designed by human annotators. This rigid structure may fail to accommodate the flexibility required to solve out-of-distribution tasks.
>   - [Robot-R1 Example]
>     - "To determine the movement needed to get to the next keypoint, we need to consider the goal of putting the rubbish into the bin …"
>     - “To determine the next waypoint for the robot to reach the button with the maroon base, we need to analyze the distance and alignment from the current position to each of the potential waypoints …”
>   - [SFT w/ CoT (MCQA) Example]
>     - “Plan: 1. Align the gripper by moving it over the base while holding the phone. …”
>     - “Plan: 1. Move the gripper to align vertically over the bin and release the gripper. …”
>
> **2. Capturing Causal Relationship**
>   - [Robot-R1]: Robot-R1 learns to generate the reasoning steps by explicitly capturing the causal relationships between them. This leads to more coherent reasoning sequences, improving overall consistency and enabling better generalization to new tasks.
>   - [SFT w/ CoT Limitation]: The training data for SFT w/ CoT is heuristically constructed, which makes it difficult to capture complex causal relationships between reasoning steps linguistically. Consequently, the diversity and quality of causal connections are inherently constrained, hindering the model’s ability to adapt to new tasks.
>   - [Robot-R1 Examples]
>     - “The task is to lift the lid off the pan using the handle. To achieve this, we need to move the robot's arm towards the pan. …”
>     - “… The coordinates relative to the robot's current position confirm that moving front and slightly left aligns with the visual information. Therefore, the next step in task planning is moving front and slightly left …”
>   - [SFT w/ CoT (MCQA) Example]
>     - “Plan: 1. Move the gripper to align vertically and position just above the red button. 2. Move the gripper down for pressing the button. Subtask: Move the gripper to align vertically and position just above the red button. …”
>     - “Plan: 1. Lift the saucepan lid upward. Subtask: Lift the saucepan lid upward. …”
>
> These elements contribute to enhanced generalization performance when encountering new tasks, and are neither included in SFT data nor easy to construct manually.
>
> ---
>
> [1] Zawalski et al., Robotic control via embodied chain-of-thought reasoning, arXiv 2024 \
> [2] Chen et al., Spatialvlm: Endowing vision-language models with spatial reasoning capabilities, CVPR 2024

---

> > ### Comment · Reviewer_5ayi · 2025-08-02
> >
> > Thank you to the author for the response. Here are my current thoughts:
> >
> > - The author may be able to improve the reasoning quality of the CoT used for SFT. The current CoT does not seem to be a true CoT but rather a mere listing of plans. Based on my experience, I believe that by putting more effort into heuristically constructing the CoT, it could be less rigid than you mentioned and even achieve better results than the spontaneous reasoning from RL that you cited.
> > - The flexibility of RL is uncontrollable. RL may produce correct answers despite erroneous reasoning, and under the RL paradigm, the agent will learn this flawed reasoning, which can hinder its reasoning ability. A heuristically constructed CoT can avoid this issue.
> > - Ensuring that RL reasoning and answers are both correct is a rather important issue.
> > - The author's response has addressed most of my questions. However, the paper indeed contains numerous terminology and formatting errors, making it somewhat difficult to read. Therefore, I will increase the score to 3 but cannot raise it higher.

---

> ### Author Response · Authors · 2025-08-05
>
> We are glad that our response has successfully addressed your concerns. To further address your remaining concern regarding CoT-SFT vs. RL, we have prepared additional discussions and experimental results.
>
>
> **1. CoT Quality Concerns**: While we agree that putting more effort into SFT dataset construction might yield improvements, we would like to emphasize that a key advantage of the RL approach is its ability to achieve better performance with comparable effort. For instance, Robot-R1 can achieve strong performance using datasets constructed with minimal domain knowledge and without external annotators, relying solely on state information. On the other hand, even our CoT-SFT dataset required more effort to incorporate human-annotated planning and subtask reasoning, yet it performed worse than Robot-R1. We expect that RL performance can be enhanced if we invest effort and resource equivalent to those used for the CoT-SFT dataset.
>
> **2. High-Quality CoT Dataset Training**: Nevertheless, to address your concerns, we try our best to create higher-quality SFT data and conduct training experiments. To this end, using GPT-4.1-mini, we generate reasoning for the MCQA data used in RL training, conditioned on human-annotated plans and subtask reasoning, with additional filtering based on correctness. As shown in the table below, there are performance gains in subtask and movement categories but still exhibited significant degradation in planning and no improvement in spatial reasoning. This demonstrates that simply improving CoT quality may not fundamentally resolve generalization and forgetting issues.
>
> **3. Complementary Relationship Between SFT and RL**: As you mentioned, reasoning from high-quality SFT models may surpass that obtained from Robot-R1. However, RL and SFT can be used together, as good SFT models provide effective initialization points for RL training (cold-start RL). To verify this, we conducted Robot-R1 training on the high quality CoT trained SFT model and achieved equal or superior performance across all tasks without performance degradation. This demonstrates the generalization capabilities of RL and the compatibility of both methodologies. The resulting reasoning would likely exceed that obtained from SFT alone.
>
> However, performance gains decreased in some tasks (spatial, movement) compared to Robot-R1 alone, presumably because SFT reduced answer diversity, constraining the RL exploration space. Training SFT with high-quality, diverse data would likely resolve this issue, which I believe is an interesting future direction to explore more.
>
> **4. RL Reasoning Reliability**: Due to RL's limited control over model’s behavior, it is possible that the model learns incorrect reasoning paths. However, experimental results show this is not severe enough to significantly impair overall generalization capabilities and performance. This issue can be mitigated through cold-start SFT approaches or reasoning path refinement and retraining using SFT.
>
>
>
> We will include the above discussion in the final draft and ensure all terminology and formatting errors are corrected.
>
>
> \begin{array}{c|cccc}
> \hline
> \text{Metric} & \text{planning} & \text{subask} & \text{movement} & \text{spatial} \newline \hline
> \text{Qwen} & 1.66 & 1.04 & 0.58 & 1.40 \newline
> \text{Robot-R1} &1.44 & 1.30 & 0.76 & 1.51 \newline  \hline
> \text{MCQA-CoT-SFT} & 1.22 & 1.28 & 0.58 & 0.37 \newline \hline
> \text{High Quality MCQA-CoT-SFT} & 0.98 & 1.34 & 0.88 & 1.4 \newline
> \text{Robot-R1 on High Quality MCQA-CoT-SFT} & 1.22 & 1.44 & 0.88 & 1.43 \newline  \hline
> \end{array}

---

> > ### Comment · Reviewer_5ayi · 2025-08-05
> >
> > Thanks. The new response has addressed my main concerns, and I will raise the score to 4.
> >
> > Additionally, I still care a lot about the consistency of the terminology used in the paper, which indeed poses a significant barrier to reading. I hope the authors can make timely corrections.

---

> > > ### Author Response · Authors · 2025-08-09
> > >
> > > We sincerely appreciate the reviewer's valuable comments and are glad to hear that our response has addressed your concerns. We will revise the terminology, adjust the formatting, and incorporate the discussion results into the final draft.

---

### Official Review · Reviewer_8V4V · 2025-06-25

**Clarity:** 2
**Significance:** 2
**Originality:** 2
**Rating:** 4
**Confidence:** 4

**Summary:**

The paper introduces ROBOT-R1 that equips LVLMs with embodied reasoning abilities tailored for robot manipulation. Expert demonstrations are converted into three multiple-choice QA tasks—predicting the next waypoint, identifying the current state, and describing the primitive movement. The LVLM must first generate an explicit reasoning trace, then select an answer; the policy is optimized with GRPO using answer correctness as reward.

**Questions:**

1. How much precision is lost by discretising the next waypoint into four options?
2. Can the final model be used in the action policy?
3. Have you sampled a subset of ROBOT-R1 Bench and compared GPT-4o scores with human annotations to quantify judge bias?
4. Real-World Validation
5. Personally, I do not think the current RLBench contained tasks have much necessities for the manipulation reasoning, since most tasks are quite straightforward and with limited steps. I would suggest try some harder benchmarks like RoboCerebra or VLABench?
6. Can this method extended to the rotation part, which is much harder to model with the language?

**Ethical Concerns:**

["NO or VERY MINOR ethics concerns only"]

**Final Justification:**

The rebuttal clears my most concerns, and I would like to keep my scores for supporting an accept.

**Limitations:**

yes

**Quality:**

2

**Strengths And Weaknesses:**

## Strengths

- Using next-state prediction into MCQA plus auxiliary tasks makes RL over continuous actions tractable and sample-efficient.

-  ROBOT-R1 Bench fills a gap by probing fine-grained embodied reasoning

## Weaknesses

- Mapping continuous keypoints to 4-option MCQA may limit precision; effect on real low-level control fidelity is unclear.
- LLM-as-judge evaluation – Reliance on GPT-4o scoring introduces potential bias; no human cross-validation.
- Simulation-only – No tests on physical robots; transfer from RLBench to reality or vision-noise robustness remains unverified.
- Reward sparsity & design – Binary correctness reward ignores partial errors in continuous space; may cause brittle policies.

---

> ### Author Rebuttal · Authors · 2025-07-31
>
> Dear Reviewer 8V4V,
>
> We sincerely appreciate your thoughtful comments. We have carefully considered each of your questions and provide detailed responses below.
>
> ---
>
> **[W1,W4,Q1] Reward sparsity & QA design**
>
> We clarify that our main goal is to enhance various embodied reasoning abilities. For establishing QA design, we compared both MCQA and open-ended QA approaches and experimentally confirmed that MCQA can better enhance embodied reasoning performance. We assume that MCQA provides more opportunities for diverse embodied reasoning through the process of comparing answer candidates, compared to open-ended approaches (see details in Table 6). Based on these results, we chose the MCQA approach despite some potential precision loss.
>
> ---
>
> **[W2,Q3] Judging bias of LLM-as-judge evaluation: Human cross-validation experiments?**
>
> To address your concern, we conducted a human cross-validation experiment on a subset of the Robot-R1 bench. See details in our response to Reviewer 5ayi's [Q6].
>
> ---
>
> **[W3] Evaluation on real robot setup**
>
> Following your constructive suggestion, we conduct an additional experiment by constructing the Robot-R1 Bench on the BridgeV2 real robot video dataset [1] and then evaluating various LVLMs, including Robot-R1 (trained on RLBench). As shown in the below table, we find that Robot-R1 consistently outperforms or is competitive with the baseline, Qwen (non-finetune), indicating Robot-R1 can be successfully transferred to real robot environments, despite the fact that it was trained using simulation data. We will include the results in the final draft.
>
> Here, one can observe that low-level control reasoning is just competitive with the non-finetuned model (Qwen), which we attribute to coordinate system changes. In RLBench, the camera position was always fixed facing the robot front, so we trained the model to describe robot direction and positional relationships based on images. With real robots, the camera position changes, requiring predictions of relative movements based on the robot arm as a reference. We believe this issue could be mitigated if the model were trained based on robot positioning from the beginning, which we plan to address in the final draft.
>
> \begin{array}{c|cccc}
> \hline
> & \textbf{Planning} & \textbf{Subtask} & \textbf{Movement} & \textbf{Spatial} \newline
> \hline
> \text{Qwen} & 1.20 & 0.89 & 0.46 & 1.53 \newline
> \text{Gemini-1.5 Pro} & 1.70 & 1.16 & 0.39 & 1.65 \newline
> \text{Robot-R1} &1.38 & 1.10 & 0.45 & 1.56 \newline
> \hline
> \end{array}
>
> ---
>
> **[Q2] Can the final model be used in the action policy?**
>
> The main goal of Robot-R1 is to enhance LVLM's embodied reasoning performance, and it would be challenging to use directly for low-level control due to insufficient precision. However, improved embodied reasoning performance often benefits VLA training [1,2], and we expect that incorporating Robot-R1 into VLA training would yield models with superior performance, which we believe is a promising direction to explore in the near future.
>
> ---
>
> **[Q5] RLBench is too simple**
>
> Thank you for your suggestion. However, we would like to clarify that RLBench contains many complex tasks that require multi-step reasoning, and we select them to construct the training and test datasets. For example, “put rubbish in bin” requests the robot agent to identify the trash, pick it up, carry it to the bin, and finally dispose of it. Furthermore, we already have considered EmbodiedBench Manipulation (see Table 4 in the paper), which includes multi-step reasoning like “Stack the front star on top of the right cylinder”. We also note that we find the improvement from Robot-R1 in more challenging environments, such as the Bridge dataset (see [W3]). Nevertheless, we plan to include more experimental results on various benchmarks that require complex reasoning in the final draft.
>
> ---
>
> **[Q6] Can this method be extended to the rotation part?**
>
> Thank you for the insightful suggestion. Although expressing rotations in language is challenging, we believe we can construct the tasks regarding rotations by using the concept of roll, pitch, and yaw. We think this extension is indeed an interesting future direction for us, and we will include relevant discussion in the final draft.
>
> ---
>
> [1] Walke et al., BridgeData V2: A Dataset for Robot Learning at Scale., Arxiv 2023
>
> ---
>
> If you have any further questions/concerns, please do not hesitate to let us know.
>
> Thank you very much,  \
> Authors

---

> > ### Comment · Reviewer_8V4V · 2025-08-05
> >
> > Q2: What's your reference in Q2, "However, improved embodied reasoning performance often benefits VLA training [1,2]"?
> >
> > And your Q2, Q5, Q6 are currently all theoretical analysis and could raises my concerns about over-claiming. Therefore, I am going to keep my score and even tending to lower my score. Please add more details.

---

> ### Author Response · Authors · 2025-08-06
>
> We first apologize for the missing references. The statement "However, improved embodied reasoning performance often benefits VLA training" corresponds to the following references: [2,3].
>
> ---
>
> **[Q2] Can the final model be used in the action policy?**
>
> To address your concern with concrete evidence, i.e., verifying whether the hypothesis actually applies to our model, we trained both Qwen2.5-VL and Robot-R1 as VLAs using the Groot-n1.5 [4] VLA training framework. We conducted experiments on the widely validated LIBERO [5] dataset for VLA training, using libero-{10, goal, object, spatial} data for 60K training steps. The results in the below table demonstrate that Robot-R1 achieved superior performance compared to the non-fine-tuned baseline model.
>
> Specifically, we observed high performance gains in Libero Goal, which requires complex operations rather than simple pick-and-place, aligning well with Robot-R1's embodied reasoning enhancements. While the Objective category showed slight performance degradation, we assume this is due to RLBench's limited object diversity during Robot-R1's training.
>
> \begin{array}{c|ccccc}
> \hline
> \text{Metric} & \text{LIBERO-10} & \text{LIBERO-Goal} & \text{LIBERO-Objective} & \text{LIBERO-Spatial} & \text{Avg}  \newline \hline
> \text{Qwen2.5 VL 7b} & 44.0 &  61.6  & 83.4  & 86.8  & 69.0  \newline  \hline
> \text{Robot-R1} & 45.8  & 73.8  & 80.4  & 87.0 & 71.8  \newline \hline
> \end{array}
>
> [2] Chen et al., Training Strategies for Efficient Embodied Reasoning, arXiv 2025 \
> [3] Yang et al., Magma: A foundation model for multimodal AI agents, CVPR 2025 \
> [4] Bjorck et al., Gr00t n1: An open foundation model for generalist humanoid robots, arXiv 2025 \
> [5] Liu et al., Libero: Benchmarking knowledge transfer for lifelong robot learning, NeurIPS 2023
>
> ---
>
> **[Q5] RLBench is too simple**
>
> As you suggested, we explored RoboCerebra, but determined it is currently unavailable as the code and required VLA for evaluation are not publicly available. Regarding VLABench,  we initially considered that it is inappropriate for Robot-R1 evaluation due to:
>
> 1) its use of multi-view images split into quadrants as a single input, along with additional annotated mask images, which differs significantly from Robot-R1's single front-view natural image training
> 2) its inclusion of reasoning elements unrelated to manipulation
>
> Nevertheless, to address your concerns, we evaluated Robot-R1 on the VLM evaluation component of VLABench. As shown in below table, performance improvements were observed in M&T (Mesh and Texture) and Spatial categories, while performance degradation occurred in PhysicalLaw and Complex categories. However, we note that PhysicalLaw and Complex are distant from our target tasks, as they require reasoning unrelated to manipulation.
>
> [PhysicalLaw task examples]
>  - "We have three objects with different densities. Choose the object with the smallest density..."
>  - "We are going to test the friction of different objects. Choose the object that slides the slowest on a steel ramp."
>
> [Complex task examples]
>  - "Please rearrange these books in chronological order starting from the first published to the latest published ones."
>  - Texas Hold'em scenario: "Secure the most formidable hand in this play."
>
> One can observe that these tasks differ significantly from Robot-R1's focus on learning movement design for task execution. Improving performance on such tasks would require joint RL training on Steam or common knowledge-related data.
> On the other hand, the categories showing improvement, particularly Spatial, are closely related to action planning and include tasks requiring multi-step manipulation reasoning. M&T contains questions involving simple manipulation.
>
> [Spatial examples]
>  - "Add 2nd condiment from bottom to the dish."
>  - "Please take the third book from the left on the top layer."
>
> [M&T examples]
>  - "Please take the book one_hundred_years_of_solitude"
>  - "Add salt to the dish"
>
> The performance improvement in Spatial demonstrates that Robot-R1 can effectively handle complex manipulation tasks despite significant environmental differences in image input. This is also significant in that it achieves performance gains while incurring minimal performance degradation, even when all SFTs fail.
>
> \begin{array}{c|cccccc}
> \hline
> \text{Metric} & \text{M\\&T} & \text{CommonSense} & \text{Semantic} & \text{Spatial} & \text{PhysicalLaw} & \text{Complex}\newline \hline
> \text{Qwen2.5 VL 7b} & 39.02 & 40.78  & 37.64 & 35.83  & 39.20 & 38.51  \newline  \hline
> \text{Robot-R1} & 46.58 & 41.5  & 37.62 & 40.33  &25.33 & 33.87  \newline \hline
> \text{CoT-SFT} & 0 & 0  & 0 & 0  & 0 & 0  \newline \hline
> \end{array}

---

> > ### Comment · Reviewer_5ayi · 2025-08-08
> >
> > Could the authors explain why the results of CoT-SFT on VLABench are entirely 0? Is the base model used Qwen2.5-vl-7B? Why is it that after the CoT-based SFT, the model is unable to complete any tasks at all?

---

> > > ### Author Response · Authors · 2025-08-09
> > >
> > > We attribute the CoT-SFT’s failure to catastrophic forgetting. This issue is caused by a fundamental mismatch between the SFT training task and the complex, agent-style requirement of the benchmark.
> > >
> > > Our CoT-SFT model was fine-tuned on Qwen2.5-VL-7B with RLBench data to take a single general image as input and perform simple QA for fixed tasks. On the other hand, VLABench’s VLM evaluation is structured as an agent-style input–output process, which, in addition to embodied reasoning, requires strong general VLM capabilities.
> > >
> > > Specifically, the model must:
> > >
> > > 1.Understand the multi-view images split into quadrants as a single image along with an object-masked image.
> > >
> > >
> > > 2.Interpret a JSON-formatted skill library via in-context learning.
> > >
> > >
> > > 3.Parse complex task requirements and generation steps.
> > >
> > >
> > > 4.Produce outputs strictly following a fixed JSON schema.
> > >
> > > This significant mismatch in the input structure between VLABench and CoT-SFT fine-tuning likely caused catastrophic forgetting of the base model’s general VLM abilities, which in turn led to failures in producing valid JSON outputs in the required format. As a result, the model was unable to complete any tasks, resulting in zero scores across the benchmark.
> > >
> > > This observation is consistent with findings from other experiments, such as Robot-R1 Bench and EmbodiedBench manipulation, suggesting that Robot-R1’s RL-based training approach has advantages over the SFT approach in mitigating forgetting and enhancing generalization.
> > >
> > >
> > >
> > > For your information, we provide an example from the VLABench prompt partition below:
> > >
> > > "... ### Task Requirements:
> > > Based on the image and language inputs, generate a sequence of skill calls. Each skill call sequence should contain the skill name (extracted from the task instruction), the skill operation parameters (if the skill requires parameters), and the target entity or container number (obtained from the labeled image). ..."
> > >
> > > "... ### Generation Steps:
> > > 1. **Extract Task Instruction**: Identify operation requirements (e.g., adjust, inspect, move) and the target object number or view from the language input.
> > > 2. **Combine with Image Information**: Using the labeled image, match the objects described in the task instruction with their corresponding identifiers to determine the target number.
> > > 3. **Generate Skill Call Sequence**: Based on the task instruction, use the extracted operation requirements as skill names and assign them the appropriate number information.
> > > 4. **Output Format**: Generate a skill call sequence in the following structure:
> > > json
> > > [
> > >     {
> > >         "name": "Skill Name 1",
> > >         "params": {
> > >             "parameter": "value"
> > >         }
> > >     },
> > >     {
> > >         "name": "Skill Name 2",
> > >         "params": {
> > >             "parameter": "value"
> > >         }
> > >     }
> > > ] …"

---

> ### Author Response · Authors · 2025-08-06
>
> **[Q6] Can this method be extended to the rotation part?**
>
> For comparison, we prepared experiments on a rotation-containing task: "take_plate_off_colored_dish_rack" as a single task. We examined performance changes when adding an auxiliary task to predict pitch, roll, and yaw for the next waypoint. Results showed that simply adding this component actually degraded performance, suggesting that predicting actions different from existing xyz movements negatively impacts existing performance. Integrating rotation would require separate research, and we believe that it would be an interesting future research direction.
> \begin{array}{c|cccc}
> \hline
> \text{Metric} & \text{planning} & \text{subask} & \text{movement} & \text{spatial} \newline \hline
> \text{Robot-R1 w/o rotation} &1.66 & 1.4 & 0.56 & 1.52 \newline  \hline
> \text{Robot-R1 w rotatotin } & 1.6 & 1.34 & 0.44 & 1.55 \newline \hline
> \end{array}
>
> We will include the above discussion in the final draft

---

> > ### Comment · Reviewer_8V4V · 2025-08-08
> >
> > Thanks for the authors effort for providing additional information.
> >
> > For the results on LIBERO, I have the following questions:
> >
> > 1. The overall performance is not that strong, especially compared with some recent open sourced SOTAs like Pi0, UniVLA, etc. The GR00t itself is a good paper but still not achieved best performance.
> >
> > 2, From your results, I am curious about why the gains are not consistent across splits, like object-split.
> >
> > Again, I would express respect towards the authors' efforts for the rebuttal phase.

---

> > > ### Author Response · Authors · 2025-08-09
> > >
> > > **[Q1] Low overall performance compared to other VLAs?**
> > >
> > > Thanks for pointing this out. This is mainly because of our experimental setup: To analyze the performance change from Robot-R1 fine-tuning, we did not use the original VLM in GR00t-n1.5 model but used Qwen and Qwen fine-tuned with Robot-R1. Unlike GR00t-n1.5’s VLM, these VLMs are not trained on large-scale cross-embodied robot datasets, but only on a small subset of RLBench data. We hypothesize that this limits the model’s capability. For your information, the original GR00t-n1.5 trained on LIBERO achieves competitive performance to other state-of-the-art VLAs.
> > >
> > > \begin{array}{c|ccccc}
> > > \hline
> > > \text{Metric} & \text{LIBERO-10} & \text{LIBERO-Goal} & \text{LIBERO-Objective} & \text{LIBERO-Spatial} & \text{Avg}  \newline \hline
> > > \text{Groot-n1.5} & 99.4 &  99.0  & 96.4  & 91.0  & 96.45  \newline  \hline
> > > \end{array}
> > >
> > >
> > >
> > > **[Q2] Object-split inconsistency**
> > >
> > > Robot-R1 was trained on RLBench, which features a limited set of objects. This likely reduced its ability to generalize to unseen objects in object-split tasks. We expect that training Robot-R1 with datasets containing a broader variety of objects, such as real-world data from BridgeV2 or simulation data from LIBERO, would mitigate this limitation.

---

> > > > ### Comment · Reviewer_8V4V · 2025-08-09
> > > >
> > > > Thanks for the authors' response and effort.
> > > >
> > > > The follow-up question:
> > > >
> > > > 1. So it seems the proposed Robot-R1 is not competitive against the original GR00t one? Yet brings some gains to the native one?  Is Qwen2.5-VL first trained with your sampled RLBench dataset, maybe not with the R1-Style, just conducting visual domain adaption with simple QA, would help or even demonstrating the similar results? Since I thought there exists huge domain gaps currently. But if Qwen2.5-VL not pre-trained with basic samples from RLBench, I thought that would be an unfair comparison. However, point out if I missed some key-information.
> > > >
> > > > 2. Okay. Reasonable. But for other splits or domains, was that less domain-gap or your Rob-R1 less sensitive?
> > > >
> > > > Due to the time limitation, no further experiments are required.

---

> ### Author Response · Authors · 2025-08-09
>
> **[Q1]**
>
> **1. Difficulty of direct comparison**
>
> GR00t-n1.5 is obtained through a multi-stage process: 1) NVIDIA trains Eagle2.5-VL[1] from Qwen2.5-Instruct on large-scale video data, and then 2) enhancing grounding capabilities and physical understanding using additional datasets. Lastly, 3) it performs VLA pre-training on robot datasets. Since Eagle2.5-VL already outperforms Qwen2.5-VL and it is undisclosed, a direct one-to-one comparison with Robot-R1 is not straightforward.
>
> For example, Robot-R1 is trained with far fewer data, about 2.5K image. However, GR00t-n1.5 uses massive embodied datasets (e.g., the OpenX Embodiment [2] dataset alone, which accounts for 27.3% of its pre-training, is around 13 TB).
>
> **2. Planned SFT comparison**
>
> To address your question, we plan to add a direct comparison experiment with an SFT-based VLA in the final draft, to show the superiority of our RL-based VLA. However, we remind you that SFT has shown limited generalization and forgetting issues compared to RL; for example, CoT-SFT was not well-generalized on VLABench unlike Robot-R1. Hence, we expect that the performance under CoT-SFT would also be limited or possibly negative.
>
> **3. Research focus and VLA integration**
>
> We first clarify that our research aims not to train VLA, but to enhance the embodied reasoning capability of VLMs. One the side, prior work such as Hi-Robot [3] has shown that generating high-level actions first, followed by hierarchical low-level action generation, can substantially boost performance. We expect that this kind of better VLA approach may fully exploit Robot-r1’s strengths under VLA framework similar to those in Pi0.5 [4] and GR-3 [5].
>
> ---
>
> **[Q2]**
>
> We believe that tasks requiring high-level action generation tend to exhibit smaller domain gaps than those relying heavily on low-level perception, since conceptual action space in text form is likely to remain more consistent across domains than raw visual input. This may explain why VLA performance improved the most on complex tasks such as LIBERO-goal and why, in Robot-R1 Bench built from BridgeV2, subtask reasoning was transferred most effectively.
>
> Overall, we really appreciate your time and effort actively spent in the discussion period. We will incorporate all of our discussions in the final draft, which we do believe would further strengthen our paper!
>
> ---
>
> [1] Chen et al., Eagle 2.5: Boosting long-context post-training for frontier vision-language models, arXiv 2025 \
> [2] O’Neill et al., Open x-embodiment: Robotic learning datasets and rt-x models, ICRA 2024 \
> [3] Shi et al., Hi robot: Open-ended instruction following with hierarchical vision-language-action models, arXiv 2025 \
> [4] Physical Intelligence et al., π₀.₅: A vision-language-action model with open-world generalization, arXiv 2025 \
> [5] Cheang et al., GR-3 Technical Report, arXiv 2025

---

> > ### Comment · Reviewer_8V4V · 2025-08-09
> >
> > Thanks for the authors' response.
> >
> > My main concern clean now. And I am going to keep my borderline accept.
> >
> > And I would encourage further experiments on VLA benchmark to demonstrate the effectiveness.

---

### Official Review · Reviewer_4LvH · 2025-07-02

**Clarity:** 3
**Significance:** 1
**Originality:** 3
**Rating:** 4
**Confidence:** 3

**Summary:**

This paper proposes Robot-R1, an LVLM training framework leveraging reinforcement learning to enhance embodied reasoning specifically for robot control. Robot-R1 is trained to use the current scene image and environment metadata to predict the next action or state. The authors propose a new benchmark to evaluate the embodied reasoning capability for Robot-R1. The results show that Robot-R1 outperforms SFT methods. The authors also evaluate Robot-R1 on the EmbodiedBench Manipulation benchmark to show the improved performance in an actual robotic agent environment.

**Questions:**

1. Could the performance of Robot-R1 be further enhanced by first performing supervised fine-tuning on a reasoning dataset as a cold start, followed by reinforcement learning (as done in DeepSeek-R1)?

2. Is it possible to observe a gradual increase in the length of generated tokens during RL training?

3. Has any decontamination been carried out to ensure that the training dataset does not overlap with Robot-R1 Bench? If so, what methodology was employed?

**Ethical Concerns:**

["NO or VERY MINOR ethics concerns only"]

**Final Justification:**

Weakness 1 only presents an analysis without statistical support. Therefore, I kept my rating.

**Limitations:**

Yes.

**Paper Formatting Concerns:**

No.

**Quality:**

2

**Strengths And Weaknesses:**

### Strengths
1. The idea of training an embodied reasoning model through reinforcement learning is interesting and worth exploring.
2. The writing of the paper is clear and easy to understand.

### Weaknesses:
1. Lack of further analysis. The paper primarily presents extensive experimental results to showcase the performance of Robot-R1, but lacks analysis on why reinforcement learning leads to performance improvements. For example, the abstract mentions that "SFT often leads to issues such as catastrophic forgetting and reduced generalization performance," yet the paper does not provide statistical evidence or concrete examples demonstrating forgetting in SFT models. It is important to show the advantages of RL training over SFT in this aspect.
Moreover, Robot-R1 is trained on multiple-choice question answering data. How does it generalize to more general embodied reasoning tasks? The latter includes planning tasks, which appear to differ significantly in distribution from the training data. Some analysis of the training data is needed to clarify this point.
2. Limitations in the evaluation. The main evaluation results of Robot-R1 are based on a benchmark proposed by the authors themselves and are evaluated using GPT-4o, which weakens the credibility of the results. Moreover, in the results shown in Table 2, Robot-R1 performs worse than GPT-4o, while in Table 3, the comparison is only made against GPT-4o-mini. A more comprehensive and systematic evaluation of Robot-R1 is necessary to support the claim in the abstract that it "surpasses GPT-4o on reasoning tasks" with greater persuasiveness.

---

> ### Author Rebuttal · Authors · 2025-07-31
>
> Dear Reviewer 4LvH,
>
> We sincerely appreciate your thoughtful comments. We have carefully considered each of your questions and provide detailed responses below.
>
> ---
>
> **[W1-1] Lack of further analysis**
>
> The evaluation benchmarks used in our paper, Robot-R1 bench, EmbodiedBench Manipulation, and SpatialRGPT, are significantly different from our training task of waypoint prediction. SpatialRGPT even uses completely different inputs based on real images. Robot-R1 demonstrates consistent performance improvements across all these environments, while SFT shows significant performance degradation in certain tasks (e.g., EmbodiedBench). This experimentally demonstrates the generalization capability of RL and the forgetting issues inherent in SFT.
>
> Furthermore, it is well-established in the LLM community that SFT causes catastrophic forgetting, while RL achieves better generalization with minimal data [1]. For instance, Deepseek-R1 showed that models trained with RL on coding and mathematics tasks also improve performance across other domains [2]. Additionally, research has demonstrated that RL can achieve generalization even with single samples [3].
>
> ---
>
> **[W1-2] RL training is good at generalization?**
>
> We would like to clarify that Robot-R1 is trained to explicitly perform embodied reasoning before answering the multiple-choice question. As a result, Robot-R1 generates various embodied reasoning, e.g., planning how to perform the task.
>
> Here, we hypothesize that wrong embodied reasoning leads to wrong responses to multiple-choice questions, and vice versa. As the model is trained to generate a correct response, we can expect that the model can generate more accurate embodied reasoning from the training. We have included the qualitative results regarding the generated embodied reasoning quality in Appendix D.
>
> ---
>
> **[W2] Limitations in the evaluation**
>
> **[Validation of Robot-R1 bench]** To address your concern, we conducted a human cross-validation experiment on a subset of the Robot-R1 bench. See details in our response to Reviewer 5ayi's [Q6]. In particular, the high correlation observed in items related to low-level control reasoning reinforces our claim that Robot-R1 "surpasses GPT-4o on reasoning tasks related to low-level action control."
>
> **[Comparison with GPT-4o]** We clarify that GPT-4o can’t be a fair baseline, as it has a significantly larger model size than Qwen-2.5 7B, which we used to train Robot-R1. As our goal is to develop the training methodology to enhance the embodied reasoning in robotics, our baseline is an initial backbone LLM (no fine-tuning) or other training methods such as SFT. For this reason, we present the performance of GPT-4o and other LLMs to provide a useful reference for better performance judgment.
>
> ---
>
> **[Q1] Cold Start SFT**
>
> We could design experiments to perform reinforcement learning on top of an SFT model that inherits Robot-R1's reasoning capabilities. We will conduct these experiments during the remaining period and aim to include the results in the final draft.
>
> ---
>
> **[Q2] Is it possible to observe a gradual increase in the length of generated tokens during RL training?**
>
> We actually observed the opposite trend that the length of the generated token is decreasing during RL training. This occurred since sentences became more concise and shifted from overview-style to descriptive formats. For specific length changes, please refer to Figure 11, and for qualitative analysis, please see Appendix C.
>
> ---
>
> **[Q3] Has any decontamination been carried out to ensure that the training dataset does not overlap with Robot-R1 Bench? If so, what methodology was employed?**
>
> Since RLBench generates new demonstrations each time, we separated the data by extracting from differently generated demos to ensure no overlap between training and evaluation datasets.
>
> ---
>
> [1] Chu et al., SFT Memorizes, RL Generalizes: A Comparative Study of Foundation Model Post-training, ICML 2025 \
> [2] Guo et al. Deepseek-r1: Incentivizing reasoning capability in llms via reinforcement learning, Arxiv 2025 \
> [3] Wang et al., Reinforcement Learning for Reasoning in Large Language Models with One Training Example, Arxiv 2025
>
> ---
>
> If you have any further questions/concerns, please do not hesitate to let us know.
>
> Thank you very much,  \
> Authors

---

> > ### Comment · Reviewer_4LvH · 2025-08-03
> >
> > Thanks for the response! Some of my concerns are addressed. However, Weakness 1 only presents an analysis without statistical support. Therefore, I kept my rating.

---

> > > ### Author Response · Authors · 2025-08-03
> > >
> > > We are glad that some of your concerns have been addressed. To further address your remaining concern, we conducted additional statistical analyses measuring performance changes with SFT and RL approaches, respectively. Specifically, for each problem type on the Robot-R1 bench, we measured the proportion of test samples showing score improvements or degradations with CoT-SFT and Robot-R1 compared to the non-fine-tuned model, Qwen-2.5 VL 7B. The results and key insights are summarized below.
> > >
> > > **Generalization**
> > >  - Robot-R1: Performance improvements via RL occur evenly across all embodied reasoning categories, including planning (18\%), subtask (36\%), and spatial reasoning (20\%) which are tasks that are completely different from the training tasks.
> > >  - CoT-SFT: Unlike RL, performance improvements via SFT with CoT in planning (2\%) and spatial reasoning (4.6\%) tasks are notably marginal, indicating limited generalization capabilities.
> > >
> > > **Forgetting**
> > >  - Robot-R1: The forgetting rate is evenly distributed across all categories and scale is relatively lower than SFT.
> > >  - CoT-SFT: Severe performance degradation is observed in certain tasks such as planning (68\%) and spatial reasoning (60\%), demonstrating catastrophic forgetting in tasks whose characteristics are far from that of the training ones.
> > > Overall, we believe that these results provide statistical evidence supporting RL's superior generalization capabilities and SFT's susceptibility to catastrophic forgetting.
> > >
> > >  \begin{array}{|l|c|c|c|c|c|} \hline
> > > \textbf{Score Improved Rate} & \textbf{Planning} & \textbf{Subtask} & \textbf{Movement} &\textbf{Spatial} & \textbf{avg} \newline
> > > \hline
> > > \text{CoT-SFT} & 2.0\\% & 36.0\\% & 24.0\\% & 4.6\\% & 15.8 \\%   \newline
> > > \hline
> > > \text{Robot-R1} & 18.0\\% & 36.0\\% & 26.0\\% & 20.0\\% & 24.7\\%  \newline \hline
> > > \end{array}
> > >
> > > \begin{array}{|l|c|c|c|c|c|} \hline
> > > \textbf{Score Decreased Rate} & \textbf{Planning} & \textbf{Subtask} & \textbf{Movement} &\textbf{Spatial} &\textbf{avg} \newline
> > > \hline
> > > \text{CoT-SFT} & 68.0\\% & 24.0\\% & 14.0\\% & 60.0\\%  & 42.8\\%   \newline
> > > \hline
> > > \text{Robot-R1} & 36.0\\% & 18.0\\% & 18.0\\% & 7.7\\% & 19.1\\%  \newline \hline
> > > \end{array}

---

> ### Author Response · Authors · 2025-08-03
>
> Furthermore, we provide intuitive examples to illustrate how Robot-R1 achieves better generalization than CoT-SFT. Specifically, we highlight two distinctive reasoning behaviors that emerged from reinforcement learning in Robot-R1, which are not observed in the CoT-SFT model
>
>
> **1. Adaptive Reasoning Planning**
>  - [Robot-R1]: Through reinforcement learning, Robot-R1 learns to adaptively determine the type and score of reasoning needed to solve the current task. This adaptive planning enhances the model to generalize effectively to new tasks.
>  - [CoT-SFT Limitation]: CoT-SFT relies on fixed, template-based reasoning patterns designed by human annotators. This rigid structure may fail to accommodate the flexibility required to solve out-of-distribution tasks.
>  - [Robot-R1 Examples]
>     - "To determine the movement needed to get to the next keypoint, we need to consider the goal of putting the rubbish into the bin …"
>     - “To determine the next waypoint for the robot to reach the button with the maroon base, we need to analyze the distance and alignment from the current position to each of the potential waypoints …”
>  - [CoT-SFT Examples]
>     - “Plan: 1. Align the gripper by moving it over the base while holding the phone. …”
>     - “Plan: 1. Move the gripper to align vertically over the bin and release the gripper. …”
>
> **2. Capturing Causal Relationship**
>  - Robot-R1: Robot-R1 learns to generate the reasoning steps by explicitly capturing the causal relationships between them. This leads to more coherent reasoning sequences, improving overall consistency and enabling better generalization to new tasks.
>  - CoT-SFT Limitation: The training data for CoT-SFT is heuristically constructed, which makes it difficult to capture complex causal relationships between reasoning steps linguistically. Consequently, the diversity and quality of causal connections are inherently constrained, hindering the model’s ability to adapt to new tasks.
>  - [Robot-R1 Examples]
>     - “The task is to lift the lid off the pan using the handle. To achieve this, we need to move the robot's arm towards the pan. …”
>     - “… The coordinates relative to the robot's current position confirm that moving front and slightly left aligns with the visual information. Therefore, the next step in task planning is moving front and slightly left …”
>  - [CoT-SFT Examples]
>     - “Plan: 1. Move the gripper to align vertically and position just above the red button. 2. Move the gripper down for pressing the button. Subtask: Move the gripper to align vertically and position just above the red button. …”
>     - “Plan: 1. Lift the saucepan lid upward. Subtask: Lift the saucepan lid upward. …”
>
> These elements contribute to enhanced generalization performance when encountering new tasks, and are neither included in SFT data nor easy to construct manually.
>
> We hope that the above statistical evidence and intuitive explanations could resolve your concerns on Robot-R1 (or RL) vs. SFT, However, if you have any other suggestions, we are also happy to incorporate during the discussion period !

---

### Official Review · Reviewer_vxzH · 2025-07-03

**Clarity:** 3
**Significance:** 3
**Originality:** 2
**Rating:** 4
**Confidence:** 3

**Summary:**

This paper introduces ROBOT-R1, a novel framework that applies Group Relative Policy Optimization to enhance embodied reasoning capabilities in Large Vision-Language Models (LVLMs) for robotic control. The key innovation lies in reformulating continuous robot control as multiple-choice question-answering tasks, where the model learns to predict next keypoint states through explicit reasoning processes. The authors train on three QA tasks: waypoint prediction, current state prediction, and primitive movement prediction, using few expert demonstrations from RLBench. Additionally, they introduce ROBOT-R1 Bench, their new benchmark for evaluating embodied reasoning in robotics.

**Questions:**

1. Can you provide results with multiple random seeds and error bars? The lack of statistical validation significantly weakens the empirical claims. How robust are your improvements across different initializations?
2. Have you tested ROBOT-R1 on real robotic systems? What are the expected challenges in transferring from simulation to reality, and how might performance degrade?
3. How does performance scale with dataset size, number of tasks, and model parameters? What are the computational requirements compared to SFT?
4. Can you provide more analysis of the failure cases of the proposed approach?
5. How did you ensure ROBOT-R1 Bench provides unbiased evaluation? Have you considered validation by human evaluators?

**Ethical Concerns:**

["NO or VERY MINOR ethics concerns only"]

**Final Justification:**

The authors satisfactorily addressed my concerns, especially the ones about extending to real robot experiments, adding multiple seeds to RL experiments, and the comparison with RL approaches apart from originally used GRPO. I believe the paper can be improved on the writing end, to add clarity, which is why I have given a final score of 4.

**Limitations:**

Please refer to the weaknesses section.

**Paper Formatting Concerns:**

None.

**Quality:**

3

**Strengths And Weaknesses:**

Strengths:
1. The paper uses DeepSeek-R1-style RL training to robotics, adapting GRPO for embodied reasoning task, where the novelty lies in reformulating continuous control as discrete multiple-choice QA, making RL tractable in high-dimensional action spaces.
2. I appreciate the comprehensive experimental evaluation across multiple benchmarks (ROBOT-R1 Bench, EmbodiedBench Manipulation, SpatialRGPT).
3. The approach shows strong performance gains on embodied reasoning tasks, outperforming GPT-4o despite having only 7B parameters (on EmbodiedBench Manipulation and SpatialRGPT). They show clear superiority over SFT baselines, which often fail completely on downstream tasks, demonstrating the efficacy of reasoning QA based approach that covers both high-level (planning, subtask) and low-level (movement, spatial) reasoning.

Weaknesses:
1. The experiments in the paper use single random seed with no error bars or statistical significance testing, which raises questions on the accuracy and reproducibility of results.
2. The results are limited to table-top manipulation tasks in simulation (RLBench), with no real robot evaluation either. The results would be stronger if tested on some other navigation or manipulation tasks.
3. The experiments use a relatively small training dataset (7.5K QA pairs across 5 tasks), which may not be enough to learn generalizable reasoning capabilities.
4. The authors use GRPO (they mention that they are inspired from DeepSeek-R1 approach). Although GRPO has clear advantages in language prediction domain, proper comparisons with actor-critic RL approaches should be done to ascertain that GRPO is indeed a good choice as an RL algorithm for embodied reasoning tasks, which is missing in the paper.
5. The approach requires access to a task-specific QA based dataset, which is not available for many embodied applications.

---

> ### Author Rebuttal · Authors · 2025-07-31
>
> Dear Reviewer vxzH ,
>
> We sincerely appreciate your thoughtful comments. We have carefully considered each of your questions and provide detailed responses below.
>
> ---
>
> **[W1,Q1] Reproducibility**
>
> We agree that online reinforcement learning can exhibit variance. To address your concern, we conducted two additional experiments with different random seeds. The results are as follows, where we confirm that our method still demonstrates significant performance improvements over the baseline. We will include this result with error bars in the final draft.
>
> \begin{array}{|l|c|c|c|c|} \hline
> \textbf{Method} & \textbf{Planning} & \textbf{Subtask} & \textbf{Primitive} &\textbf{Spatial} \newline
> \hline
> \text{Qwen} & 1.66 & 1.04 & 0.58 & 1.40 \newline
> \text{GRPO seed=1} & 1.44 & 1.30 & 0.76 & 1.51 \newline
> \text{GRPO seed=2} & 1.70 & 1.46 & 0.64 & 1.66 \newline
> \text{GRPO seed=3} & 1.56 & 1.50 & 0.54 & 1.60 \newline
> \text{GRPO avg} & 1.57 & 1.42 & 0.65 & 1.59 \newline
> \hline
> \text{RLOO} & 1.56 & 1.54 & 0.68 & 1.52 \newline
> \text{Reinforce++} & 1.40 & 1.08 & 0.62 & 1.38 \newline \hline
> \text{GRPO with Half dataset} & 1.56 & 1.50 & 0.54 & 1.60 \newline
> \text{GRPO with single task} & 1.52 & 1.28 & 0.54 & 1.58 \newline \hline
> \text{Qwen 3B} & 1.66 & 1.04 & 0.58 & 1.40 \newline
> \text{Qwen 3B + Robot-r1} & 1.58 & 1 & 1.08 & 1.23 \newline\hline
> \end{array}
>
> ---
>
> **[W2,Q2] Evaluation on diverse tasks and real robot system**
>
> To address these concerns, we created a new Robot-R1 bench based on BridgeV2 real robot data and evaluated our model's performance on it. The results showed that performance improvements remained prominent across high-level actions, indicating Robot-R1 can be successfully transferred to real robot environments, despite the fact that it was trained using simulation data, RLBench. We will include the results in the final draft.
>
> Here, one can observe that low-level control reasoning did not transfer well, which we attribute to coordinate system changes. In RLBench, the camera position was always fixed facing the robot front, so we trained the model to describe robot direction and positional relationships based on images. With real robots, the camera position changed, requiring predictions of relative movements based on the robot arm as a reference. We believe this issue could be mitigated if the model were trained based on robot positioning from the beginning, which we plan to address in the final draft.
>
> \begin{array}{c|cccc}
> \hline
> & \textbf{Planning} & \textbf{Subtask} & \textbf{Primitive} & \textbf{Spatial} \newline \hline
> \text{Qwen} & 1.20 & 0.89 & 0.46 & 1.53 \newline
> \text{Gemini-1.5 Pro} & 1.70 & 1.16 & 0.39 & 1.65 \newline
> \text{Robot-R1} &1.38 & 1.10 & 0.45 & 1.56 \newline
> \hline
> \end{array}
>
> ---
>
> **[W3] Dataset size issue**
>
> We would like to clarify that models trained with reasoning incentivized RL can generalize effectively with less than 10K training data [1,2], and a recent study has even shown that generalization is possible from a single example [3].
>
> In addition, we find that Robot-R1 trained with a 7.5K QA dataset shows consistent performance gains across diverse tasks from different benchmarks (Robot-R1 Bench, EmbodiedBench Manipulation, and SpatialRGPT-Bench), which empirically demonstrates that the 7.5K QA dataset is sufficient to learn generalizable reasoning capability.
>
> ---
>
> **[W4] Traning algorithm comparison**
>
> We agree with the reviewer in that investigating the proper RL algorithms for embodied reasoning tasks is an interesting problem. However, actor-critic RL like PPO requires excessive GPU VRAM and hence it’s hard to apply in the current constrained resources. Instead, we conduct an experiment by training Robot-R1 with widely used RL algorithms, RLOO [4] and Reinforce++ [5].
>
> As shown in the above table in [W1,Q1], Robot-R1 trained with RLOO achieves similar performance to Robot-R1 trained with GRPO, and Reinforce++ degrades the performance. We think this is because the unstable training of Robot-R1 with Reinforce++, due to weaker reward normalization (batch reward normalization).
>
> We will include this discussion in the final draft.
>
> ---
>
> **[W5] The approach requires access to a task-specific QA based dataset, which is not available for many embodied applications.**
>
> The main goal of Robot-R1 is to enhance VLM's embodied reasoning performance for robotic applications. The training data is constructed based on expert demonstration state trajectories, which can be readily obtained from most robotic learning datasets.
>
> These reasoning abilities can be easily transferred to various tasks, including EmbodiedBench, real robot experiments [W2, Q2], and other benchmarks. This experimentally demonstrates that our approach is not heavily constrained by task-specific datasets.
>
> ---
>
> **[Q3-1] Scaling law: How does performance scale with dataset size, number of tasks, and model parameters?**
>
> We conducted experiments involving three modifications: reducing the training data size by half, utilizing a single task (put rubbish in bin) for training, and downsizing the model from 3B parameters.
>
> As shown in the above table in [W1,Q1], the results demonstrated that Robot-R1 generalizes effectively even with smaller models. When the data volume was reduced by half, performance remained comparable within the margin of error. However, when limiting the number of tasks to single, we observed a slight performance degradation in subtasks. As mentioned in [W3], we attribute these results to the robust generalization capabilities of RL training.
>
> ---
>
> **[Q3-2] Computing cost: What are the computational requirements compared to SFT?**
>
> Training a model with RL (GRPO) requires around 1.5 times more training time than SFT.
>
> ---
>
> **[Q4] Can you provide more analysis of the failure cases of the proposed approach?**
>
> As mentioned in [W2,Q2], the low-level control reasoning capabilities learned from RLBench showed limited transferability to other robotic systems. This limitation is presumed to arise from the discrepancy between the coordinate systems used during the training phase and those employed during the testing phase.
>
> ---
>
> **[Q5] Judging bias of LLM-as-judge evaluation: Human cross-validation experiments?**
>
> Thank you for pointing this out. To address your concern, we conduct a human cross-validation experiment on a subset of the Robot-R1 bench. See details in our response to Reviewer 5ayi's [Q6].
>
> ---
>
> [1] Zeng et al., Simplerl-zoo: Investigating and taming zero reinforcement learning for open base models in the wild, Arxiv 2025 \
> [2] LMMS-Lab, Multimodal Open R1 8K Verified, Hugging Face, 2024/2025 \
> [4] Wang et al., Reinforcement Learning for Reasoning in Large Language Models with One Training Example, Arxiv 2025 \
> [5] Ahmadian et al., Back to basics: Revisiting reinforce style optimization for learning from human feedback in llms, Arxiv 2024 \
> [6] Hu et al., Reinforce++: An efficient rlhf algorithm with robustness to both prompt and reward models, Arxiv 2025
>
> ---
>
> If you have any further questions/concerns, please do not hesitate to let us know.
>
> Thank you very much,  \
> Authors

---

> > ### Comment · Reviewer_vxzH · 2025-08-06
> >
> > I appreciate the additional RL-based experiments and the inclusion of multiple seeds. I also value the real robotic experiments on BridgeV2, which I believe strengthen the paper's claims. My concerns have been satisfactorily addressed, and I will maintain my prior assessment of the paper. Please ensure the extra experiments and details are included in the draft.

---

> > > ### Author Response · Authors · 2025-08-09
> > >
> > > We are glad to hear that your concerns have been addressed. All the content discussed will be included in the final draft of the paper. We sincerely thank you again for your valuable comments and feedback.

---

### Note · Authors · 2025-08-13

Dear Reviewers and Area Chair,

We first thank all reviewers for their positive scores and feedback and sincerely appreciate your time and effort in serving the Neurips community.

As highlighted by the reviewers, we present a reinforcement learning framework that improves embodied reasoning (vxzH, 4LvH). Specifically, without relying on costly embodied task-specific label annotations, Robot-r1 effectively leverages the robot’s numeric state, which is inexpensive and easy to obtain, to enable a simple and practical implementation (vxzH, 8V4V, 5ayi). Furthermore, we introduce Robot-R1 Bench, a benchmark designed to enable precise evaluation of fine-grained embodied reasoning, addressing a gap in existing benchmarks (8V4V). Evaluations on Robot-R1 Bench and diverse benchmarks show that our method consistently surpasses SFT and other baselines, while achieving higher sample efficiency and stronger generalization (vxzH, 5ayi).

We are delighted to have addressed the common concern regarding the validation of Robot-R1 Bench (vxzH, 4LvH) by providing new human cross-validation results. We also responded to requests for evaluation in more complex environments by presenting additional results on the VLABench and on a real-robot benchmark based on BridgeV2.

For reviewer 5ayi, who expressed concerns about novelty and comparisons with the SFT baseline, we clarified the unique aspects of embodied settings, and providing additional experimental results using high-quality SFT CoT data. As a result, the reviewer’s concerns were successfully resolved as acknowledged by the increase of score from 2 to 4.

The remaining reviewers initially provided positive feedback, and our responses during the discussion phase further strengthened their positive recognition of our work. To be specific, for reviewer vxzH, we presented results demonstrating consistent performance across multiple seeds and comparisons with other RL algorithms. For reviewer 4LvH, we provided additional explanations on why Robot-R1 outperforms SFT, supported by both qualitative and quantitative evidence as well as references to relevant literature. For reviewer 8V4V, we showed that VLA training offers tangible benefits for real-world action generation.


We are grateful for the valuable feedback, which allowed us to strengthen the credibility of our work. We believe this work will enable more cost-effective and scalable research in embodied AI, making a positive impact on the NeurIPS community.

Authors

---

### Decision · Program_Chairs · 2025-09-17

**Decision:**

Accept (poster)

**Comment:**

[Summary]

The paper proposes ROBOT-R1, a reinforcement learning (RL) framework to enhance embodied reasoning in Large Vision-Language Models (LVLMs) for robot control. The key idea is to reformulate continuous control into multiple-choice QA tasks (e.g., waypoint prediction, state prediction, primitive movement), enabling tractable RL optimization with Group Relative Policy Optimization (GRPO). The authors further introduce ROBOT-R1 Bench, a benchmark for embodied reasoning evaluation. Empirical results show consistent gains over SFT baselines and competitive performance with much larger closed-source models across several settings (ROBOT-R1 Bench, EmbodiedBench Manipulation, SpatialRGPT). Additional rebuttal experiments address reproducibility, RL algorithm comparisons, and transfer to real-robot datasets (BridgeV2).

[Strengths]

(1) Novel training reformulation: Casting robot control as multiple-choice QA is an elegant way to apply RL effectively in high-dimensional action spaces.
(2) Empirical effectiveness: ROBOT-R1 demonstrates clear improvements over SFT baselines, particularly in embodied reasoning tasks where SFT often struggles with catastrophic forgetting and poor generalization.
(3) Breadth of evaluation: Results span multiple benchmarks (including new and existing ones), with further validation on real robot data (BridgeV2). This strengthens claims of transferability beyond RLBench simulation.
(4) Thorough rebuttal engagement: Authors provided convincing additional experiments: multi-seed results, RL algorithm ablations (GRPO vs RLOO vs Reinforce++), scaling-law analysis, human cross-validation of LLM-as-judge, and preliminary VLA integration (via GR00t/LIBERO). These responses addressed the reviewers’ major methodological concerns.
(5) Clarity of contribution: Despite some terminology issues, the overall pipeline and design are relatively easy to understand, making this work accessible to a broad robotics + multimodal AI audience.

[Weaknesses]

(1) Limited scope of evaluation: Most core experiments remain restricted to tabletop manipulation in simulation. Although real-robot validation was added, low-level control transfer is still weak. Stronger evidence on diverse robotics domains (e.g., navigation, multi-step real-world manipulation) would further substantiate the claims.
(2) Benchmark credibility: Heavy reliance on an author-proposed benchmark (ROBOT-R1 Bench) and LLM-as-judge evaluation may raise concerns of bias. Human validation was conducted, but scope remains limited.
(3) Terminology and consistency issues: Multiple reviewers noted confusing or inconsistent definitions of “planning,” “subtask,” “primitive movement,” etc., which hinder readability.
(4) Comparisons: While the method improves upon SFT, direct comparisons against stronger RLHF/VLA baselines (e.g., Pi0, UniVLA, GR00t) are missing or remain preliminary. This makes it harder to position ROBOT-R1 against recent state-of-the-art.
(5) Limited gain of applying it for real robot manipulation tasks. There is clearly a gap of improving the performance on the proposed embodied reasoning benchmark and the final performance on the end-to-end manipulation tasks as indicated by the experiment results shared by the authors.

[Reviews]

All four reviewers gave borderline accept (score 4) after rebuttal. Their initial concerns like variance across seeds, lack of real-robot validation, weak comparison to other RL algorithms, and limited analysis, were largely addressed through additional experiments and clarifications. Remaining criticisms mainly concern clarity, limited scope, and over-claiming relative to SOTA. Importantly, no reviewer advocated rejection after rebuttal; instead, they acknowledged the paper’s technical soundness and the significance of its contribution to embodied reasoning in LVLMs.

[Conclusion]

While the ACs still hold some concerns about this work, this work presents an interesting idea of integrating RL into embodied multimodal reasoning. The ACs thus recommend an acceptance, but strongly recommend the authors to incorporate the discussions, related works and newly added experiments into the final version. Moreover, the authors should clearly show the transferability of performance gain on the proposed benchmark to the real robot manipulation tasks in their final version.